# Neofunctionalization of an ancient domain allows parasites to avoid intraspecific competition by manipulating host behaviour

Jiani Chen[1,6], Gangqi Fang [2,3,6], Lan Pang[1], Yifeng Sheng[1], Qichao Zhang[1], Yuenan Zhou[1], Sicong Zhou [1], Yueqi Lu[1], Zhiguo Liu[1,4], Yixiang Zhang[2,3], Guiyun Li[2], Min Shi[1,4], Xuexin Chen [1,4,5✉], Shuai Zhan [2,3✉] & Jianhua Huang [1,4✉]

Intraspecific competition is a major force in mediating population dynamics, fuelling adaptation, and potentially leading to evolutionary diversification. Among the evolutionary arms races between parasites, one of the most fundamental and intriguing behavioural adaptations and counter-adaptations are superparasitism and superparasitism avoidance. However, the underlying mechanisms and ecological contexts of these phenomena remain underexplored. Here, we apply the *Drosophila* parasite *Leptopilina boulardi* as a study system and find that this solitary endoparasitic wasp provokes a host escape response for superparasitism avoidance. We combine multi-omics and in vivo functional studies to characterize a small set of RhoGAP domain-containing genes that mediate the parasite's manipulation of host escape behaviour by inducing reactive oxygen species in the host central nervous system. We further uncover an evolutionary scenario in which neofunctionalization and specialization gave rise to the novel role of RhoGAP domain in avoiding superparasitism, with an ancestral origin prior to the divergence between *Leptopilina* specialist and generalist species. Our study suggests that superparasitism avoidance is adaptive for a parasite and adds to our understanding of how the molecular manipulation of host behaviour has evolved in this system.

[1] Institute of Insect Sciences, Ministry of Agriculture Key Lab of Molecular Biology of Crop Pathogens and Insect Pests, College of Agriculture and Biotechnology, Zhejiang University, Hangzhou, China. [2] CAS Key Laboratory of Insect Developmental and Evolutionary Biology, CAS Center for Excellence in Molecular Plant Sciences, Chinese Academy of Sciences, Shanghai, China. [3] CAS Center for Excellence in Biotic Interactions, University of Chinese Academy of Sciences, Beijing, China. [4] Key Laboratory of Biology of Crop Pathogens and Insects of Zhejiang Province, Zhejiang University, Hangzhou, China. [5] State Key Lab of Rice Biology, Zhejiang University, Hangzhou, China. [6] These authors contributed equally: Jiani Chen, Gangqi Fang. ✉email: xxchen@zju.edu.cn; szhan@sibs.ac.cn; jhhuang@zju.edu.cn

How competition shapes the communities of animals and plants has been one of the central topics in ecology and evolution for decades[1–5]. Under limited resources, intraspecific competition limits the increase in population size and potentially lowers genetic diversity due to genetic drift; hence, the population is more likely to become extinct when exposed to new environments[4,6,7]. On the other hand, competition induces natural selection that might enable rapid adaptation to novel conditions and niche expansion for the populations at longer timescales[5,8,9]. Intraspecific competition is widespread and particularly important in communities of parasitoid wasps and other parasites, whose development closely relies on the condition of hosts[10]. Furthermore, the offspring of parasitic wasps are typically confined to the host that was parasitized, so the strategies of the ovipositing female therefore largely determine her reproductive outputs; her strategy is therefore under strong natural selection.[11] Superparasitism, i.e. two or more eggs being laid into a single host by one or more parasitoid females, and superparasitism avoidance are among the most fundamental intraspecific competition behaviours of solitary wasps[12–14]. However, issues regarding their evolutionary consequences and fitness costs have long been debated[12,15–17].

In most conditions, parasitoid females refuse to lay additional eggs into already parasitized hosts (superparasitism avoidance), since superparasitism would introduce intense competition between wasps in a narrow niche[12,18]. Superparasitism seriously lowers the reproductive efficiency of parasitoids by affecting the body size or weight of parasitoid offspring, extending the developmental time of wasp progeny, and increasing premature mortality of both developing hosts and parasitoids[19–23]. Despite being considered disadvantageous, female parasitoids superparasitize on some occasions. This paradoxical decision was first interpreted as being the consequence of 'errors' on the part of the egg-laying females but was later re-interpreted as an adaptive decision when females have no better solution, i.e. when the overall environmental quality is low[12,24,25]. Furthermore, recent studies discovered an inheritable filamentous virus (FV) that can manipulate superparasitism in wasps, suggesting that superparasitism is likely to be manipulated by symbionts as an adaptive strategy for virus transmission[13,26,27]. Despite long being of interest and the topic of debate, the detailed mechanism underlying superparasitism avoidance remains largely underexplored, which limits further addressing its enigmatic ecological context. *Drosophila* parasitoids have been widely utilised as both experimental and theoretical models to study the mechanisms underpinning the host response to parasite infection[28–34].

In this study, we use the solitary endoparasitoid *Leptopilina boulardi* (Lb) as the study system to characterise the molecular basis underlying the avoidance of parasitoid superparasitism. We find that Lb avoids superparasitism by manipulating the host escape behaviour via inducing reactive oxygen species in the host central nervous system. By tracing the evolutionary scenario of this mechanism, we find that the neofunctionalization and expression specialisation gave rise to the novel role of an ancient RhoGAP domain which allows parasitoids to manipulating the host behaviour.

## Results

**Parasitisation of Lb induces host escape behaviour.** Solitary parasitoids usually lay a single egg inside the host per each oviposition. Even when multiple ovipositions occur, at most one of the laid offsprings can successfully develop into the adult and emerge from a single host (if survived)[10–12]. Thus, behavioural qualification of "superparasitism" in solitary parasitoids is unambiguous, i.e., the appearance of more than one egg in a

single host. This strictly exclusive pattern makes this type of parasitoids as an ideal model to study superparasitism. Lb is a classic solitary parasitoid of *Drosophila* that mainly attacks 2nd instar larvae of *D. melanogaster*, and adult wasps emerge when hosts are at the pupal stage[28]. The Lb line in this study was checked to verify the absence of LbFV infection ('Methods'), and its parasitisation strategy was observed under different parasite to host ratios. When fly larvae were exposed to 3-day-old Lb females with a parasite to host ratio of 1:10 in standard fly food bottles, we found that most larvae contained a single (parasitoid) egg after 60 min, while the prevalence of hosts with two or more eggs increased substantially thereafter to become almost universal by 240 min (Fig. 1a). Similarly, the parasitized *Drosophila* hosts generally had only one Lb egg during the early phase of parasitism, but two or more eggs per host were often observed when wasps were present for a long time when there was a 1:20 parasite:host ratio (Fig. 1b). These results confirm that Lb usually has a tendency to avoid superparasitism when there are sufficient numbers of unparasitized hosts.

Interestingly, when we observed the parasitism process of Lb, the very frequent escape of hosts, i.e., *D. melanogaster* larvae, attracted our attention. After the release of parasitoids, some *Drosophila* larvae withdrew from their food and crawled back and forth between the food surface and the walls of the bottles (Supplementary Movies 1 and 2). Parasitoid wasps prefer to parasitise hosts on food (Supplementary Movie 2). Escape from the food of hosts led us to speculate that such escape behaviour may have been induced by the parasitism of a parasitoid that tries to manipulate the hosts' behaviour to avoid superparasitism, or simply indicates that larvae try to escape from being attacked by parasitoids. To differentiate these two hypotheses, it is important to test whether the escaped hosts had been parasitized.

We first tested whether the parasitism of Lb can effectively induce host escape and, consequently, contribute to superparasitism avoidance. Under our standard conditions of a 1:10 parasite:host ratio, we observed evident escape behaviour in the host larvae (Fig. 1c). This response was largely initiated 15 min after parasitoids were introduced and was quantified by removing larvae from the bottle wall during the experiment (not allowing any to return to the food) and summing the collected larvae at 15, 30, 45 min and so forth (Fig. 1d). Detailed observations showed that this host escape behaviour lasted ~3 h when the wasps were left in the bottle for the entire 4-h period (Fig. 1d). In total, we found that more than 77% of the *Drosophila* larvae showed escape behaviour in the assay. This altered behaviour was not observed if no wasps were introduced (Fig. 1d). As expected, the majority (74.34%) of non-escaped hosts died in the larval or pupal stage without yielding adult flies or wasps (Fig. 1e and Supplementary Fig. 1). We then collected and dissected host larvae that moved to the bottle wall and found that more than 95% of these 'escaped' larvae contained at least one wasp egg (Fig. 1f and Supplementary Fig. 2a). In contrast, the non-escaped *Drosophila* larvae contained many more eggs than the ones that had shown escape behaviour (one-way ANOVA, $P < 0.0001$; Supplementary Fig. 2a), supporting that eventually the non-escaping hosts suffered higher levels of superparasitism. Thus, these experiments reveal that the parasitisation of Lb triggers the host to escape from its food, which might help this solitary parasitoid avoid superparasitism.

**Venom is required to induce the escape behaviour of hosts.** Interestingly, we did not find frequent host escape if only male wasps were introduced (Fig. 1d), i.e., only female Lb can induce host escape. We hypothesise that female-specific venom probably plays an important role in inducing host escape because it is co-injected with wasp egg(s) during oviposition. In the previous 4-h

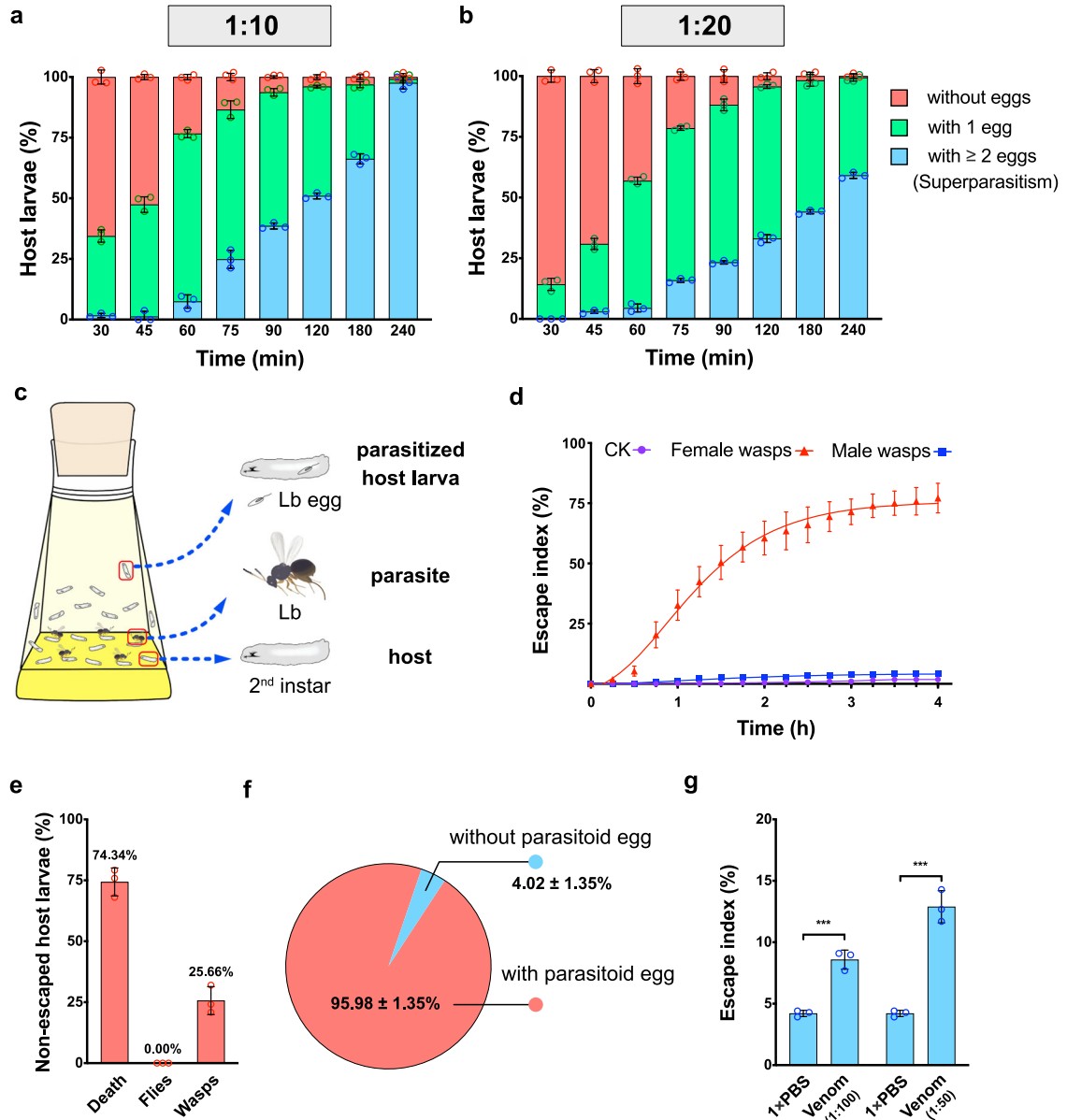

**Fig. 1 Parasitoid venom is necessary for host escape behaviour. a** Percentage of host larvae containing the indicated numbers of wasp eggs after exposure to female Lb at a parasite to host ratio of 1:10 in standard fly food bottles for 30 min ($n = 245$), 45 min ($n = 215$), 60 min ($n = 285$), 75 min ($n = 298$), 90 min ($n = 256$), 120 min ($n = 226$), 180 min ($n = 157$) or 240 min ($n = 209$). Three biological replicates were performed. Data are presented as mean values ± SD. **b** Percentage of host larvae containing the indicated numbers of wasp eggs after exposure to female Lb at parasite to host ratios of 1:20 for 30 min ($n = 146$), 45 min ($n = 209$), 60 min ($n = 240$), 75 min ($n = 201$), 90 min ($n = 194$), 120 min ($n = 166$), 180 min ($n = 156$) or 240 min ($n = 178$). Three biological replicates were performed. Data are presented as mean values ± SD. **c** Schematic of the experiment in which wild-type *Drosophila* 2nd instar (host) larvae in fly bottles were exposed to Lb female wasps. The images were crafted using Affinity Designer v1.8.6 and Procreate v4.2.2. **d** The escape indices (number of hosts exhibiting short-term escape behaviour divided by the total number of hosts multiplied by 100) of hosts exposed to no wasp control (CK, purple curve, $n = 380$), to female wasps for 4 h (female wasps, red curve, $n = 358$) and to male wasps (male wasps, blue curve, $n = 461$). The number of escaped hosts was recorded and summed every 15 min. Three biological replicates were performed. The difference in escape index between treatments was determined by one-way ANOVA along with Fisher's least significant difference test (see statistics of each time point in Supplementary Table 4). **e** The percentage of remaining host larvae that died, emerged as flies and hatched wasps after exposure to wasps for 4 h. Data are presented as mean values ± SD. **f** The percentage of host larvae exhibiting escape behaviour that contain Lb eggs in behaviour assays. **g** Escape indices for *Drosophila* hosts treated with venom at 1:100 and 1:50 dilutions, with 1× PBS as controls. Left to right: $n = 240$, 296, 240 and 263. At least three biological replicates were performed. Data are presented as mean values ± SD. Significance was determined by two-sided unpaired Student's *t* test (1:100 venom: $P = 0.0008$; 1:50 venom: $P = 0.0004$; ***$P < 0.001$). Source data are provided as a Source data file.

assay, we found that most parasitized host larvae presented escape behaviour from 30 to 75 min after cohabitation with female wasps (Supplementary Fig. 2b). To test whether venom is responsible for manipulating the host behaviour, we thus injected *D. melanogaster* 2nd instar larvae with 1:100 or 1:50 dilutions of the total

venom extract derived from a single female wasp and analysed their escape performance from 30 to 75 min after injection (Fig. 1g). Injection of each individual dose markedly initiated the host escape response compared to that of the 1× PBS-treated control. Moreover, the escape index of venom-injected host larvae

**a**

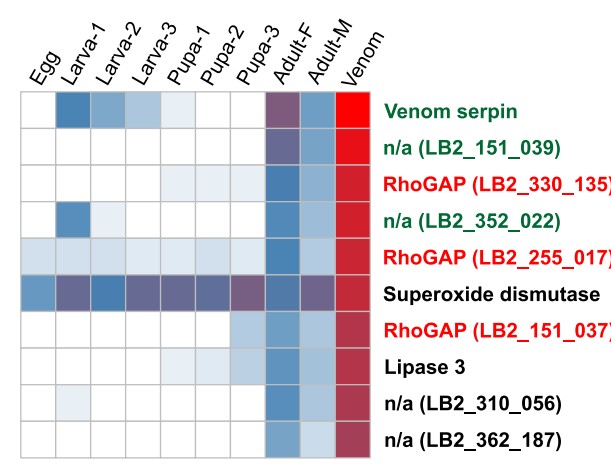

**b**

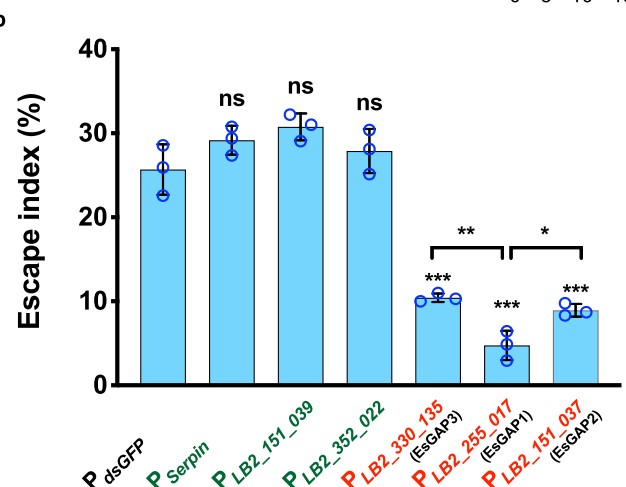

**Fig. 2 A family of EsGAP venom proteins contributes to escape behaviour. a** Transcription heatmap of all venom-protein genes that were significantly highly expressed in venom. Genes are displayed in order of expression levels in the venom. The three genes in red were predicted to have RhoGAP domains and were subjected to functional testing together with another three highly expressed genes (in green). **b** Escape indices of host larvae that were parasitized by Lb with knockdown of the indicated proteins using an RNAi strategy. dsGFP-treated Lb (P$_{dsGFP}$) was used as a control. Drosophila larvae infected by parasitoids with knockdown of EsGAP had the lowest escape index. The number of larvae exhibiting escape behaviour was recorded from 30 min to 75 min after exposure to Lb. Left to right: n = 360, 278, 323, 424, 317, 391 and 316 biologically independent host larvae. Three biological replicates were performed. Data are presented as mean values ± SD. Significance was determined by two-sided unpaired Student's t test (P$_{Serpin}$: P = 0.1564; P$_{LB2\_151\_039}$: P = 0.0609; P$_{LB2\_352\_022}$: P = 0.3953; P$_{EsGAP3}$: P = 0.001; P$_{EsGAP1}$: P = 0.0005; P$_{EsGAP2}$: P = 0.0007; P$_{EsGAP3}$ vs P$_{EsGAP1}$: P = 0.0056; P$_{EsGAP2}$ vs P$_{EsGAP1}$: P = 0.0188; *P < 0.05; **P < 0.01; ***P < 0.001; ns: not significant). Source data are provided as a Source data file.

was significantly higher at a 1:50 dilution than at a 1:100 dilution (Fig. 1g). Thus, our results suggest that Lb parasitisation triggers host escape behaviour mainly via the co-injected venom.

**Identification of the full-venom-protein catalogue of Lb.** What component of venom plays an essential role in manipulating host behaviours? Venom generally undergoes rapid evolution, making the gene repertoire of venom dramatically varied even between

related species and difficult to precisely and completely annotate[35]. To obtain a complete catalogue of Lb venom proteins (VPs), we first generated an improved Lb genome by de novo assembling long sequencing reads ('Methods'). The presented 354.8 Mb reference genome was of high contiguity (N50 size: 2.7 Mb) and of high completeness (Supplementary Table 1). We further sampled venom fluid from the female Lb venom reservoir and sequenced the proteome of Lb venom in combination with the transcriptome of the venom gland. Of 12,613 protein-coding genes ('Methods'), we characterised 91 genes as reliable venom proteins that were greatly supported by both transcriptomic and proteomic evidence (Supplementary Table 2; 'Methods'). Most of these VP genes were either species-specific (29.7%) or annotated as having an uncharacterised function (28.6%); none of the known gene families or pathways was significantly enriched in the set. We thus gave priority to those highly expressed in the venom. Of 91 VP genes, 10 were extremely highly expressed in the venom (Z test; P < 0.01), including four species-specific genes and, remarkably, three RhoGAP domain-containing genes (Supplementary Table 2). The three RhoGAP genes shared a close gene length and all encoded a GTPase activating protein domain (PF00620). More importantly, they all showed remarkable expression specialisation in venom (Fig. 2a). These features led us to propose an unusual role of the RhoGAP domain in the venom.

**Knockdown of RhoGAP genes in Lb deprives escape behaviour in hosts.** We next performed in vivo knockdown experiments to explore the functional relationships of these highly expressed VPs with host escape. All three RhoGAP genes (LB2_330_135, LB2_255_017, and LB2_151_037) and the other three most highly expressed genes (LB2_216_018, a putative serpin gene; LB2_151_039 and LB2_352_022, Lb-specific genes) were selected for RNA interference (RNAi) experiments in Lb (Supplementary Fig. 3a). The qRT-PCR results showed that the expression levels of these genes were all significantly reduced in Lb adults after injection of the corresponding double-stranded RNA (dsRNA) into fifth instar wasp larvae (Supplementary Fig. 3b), but we noticed the non-specific reduced expression across the three RhoGAP genes, possibly due to the sequence similarity between each other. The resulting dsRNA-treated Lb females were used to observe host-larva escape behaviour. We found that knockdown of the three RhoGAP genes all results in a significantly lower escape index in hosts, with all greater than 50% effect, in comparison to that of the control group and those with knockdown of other tested genes (Fig. 2b). Despite the spillover effect, the suppressing effect of host escape behaviour by LB2_255_017 was much stronger than those of the other two RhoGAP genes (i.e., LB2_330_135 and LB2_151_037; Fig. 2b) in the assay. We termed these genes Escape-related genes with a GTPase Activating Protein domain (EsGAP) due to their functional relationship with host escape, and named these genes EsGAP1-3, respectively, based on their observed effects on inducing host escape (Fig. 2b).

**Parasitisation induces the accumulation of ROS in hosts.** How do these parasitoid EsGAP genes manipulate the behaviour of hosts? RhoGAP is the activating protein of GTPase, which has been widely suggested to participate in cytoskeletal regulation, cell proliferation, and reactive oxygen species (ROS) production[36–40]. Correspondingly, Drosophila larvae exposed to hypoxia are known to show markedly increased levels of ROS in the central nervous system (CNS) and to crawl away from food[41]. We, therefore, examined whether ROS levels were induced in hosts upon parasitisation. Escaping larvae were dissected 60 min after exposure to wasps and stained with the ROS detector 2′,7′-dichlorofluorescein diacetate (DCFH-DA). Interestingly, the host

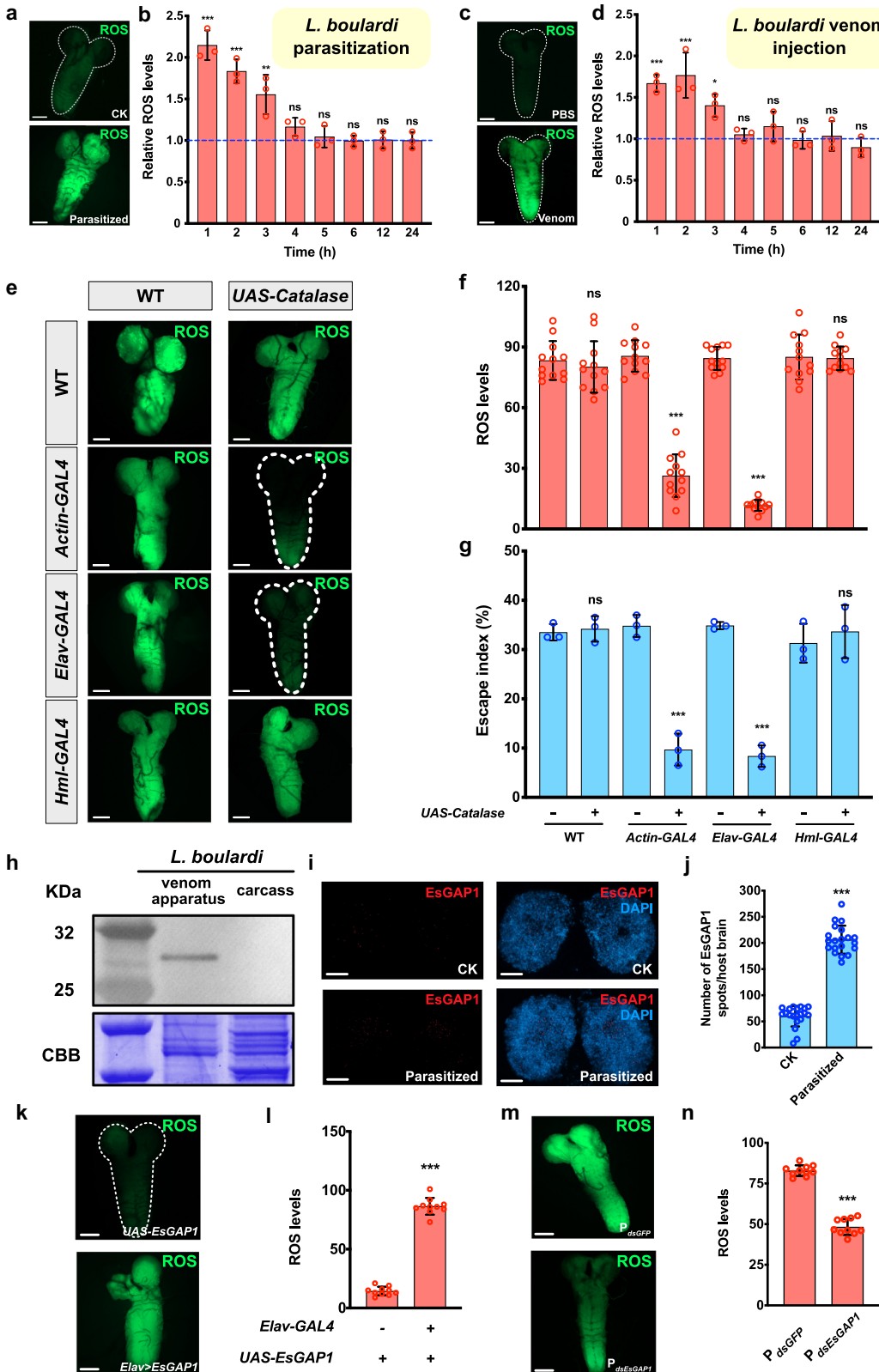

CNS showed strongly elevated ROS levels compared to those of normal larvae (Fig. 3a), while most other tissues appeared indistinguishable between parasitized and nonparasitized larvae (Supplementary Fig. 4, see details of other tissues in 'Methods'). Thus, the consequent escape behaviour of hosts was likely to be induced by the increase in ROS in the CNS. Moreover, when we evaluated ROS levels at different time points after Lb infection, the ROS levels in parasitized host CNSs were significantly higher during the first 3 h after parasitisation and then declined back to normal in comparison to the CNS of nonparasitized larvae. This changing pattern is consistent with that of the behavioural response of parasitized hosts, i.e. hosts mainly escape during the first 3 h after exposure to parasitoids (Fig. 3b and Supplementary Fig. 2b).

**Fig. 3 ROS levels are induced by parasitisation and important for escape behaviour. a** Representative images of the CNSs of nonparasitized control (CK) and Lb-parasitized larvae (Parasitized). The CNS was stained with DCFH-DA. Scale bars: 50 μm. **b** ROS levels as a function of time in larval CNS after exposure to female Lb wasps ($n = 3$ replicates, at least seven *Drosophila* CNSs were examined for each individual). The dashed line indicates the ROS levels of the nonparasitized larvae, which was used as a control. Data are presented as mean values ± SD, and significance was analysed using one-way ANOVA followed by Šidák's multiple-comparison test (1 h: $P = 6.2 \times 10^{-8}$; 2 h: $P = 6.5 \times 10^{-6}$; 3 h: $P = 0.001$; 4 h: $P = 0.7629$; 5 h: $P = 0.9999$; 6 h: $P = 0.9999$; 12 h: $P = 0.9999$; 24 h: $P = 0.9999$; **$P < 0.01$; ***$P < 0.001$; ns: not significant). **c** Representative images of CNSs from 1× PBS-injected control (PBS) and Lb venom (1:50 dilution)-injected larvae (Venom). The CNS was stained with DCFH-DA. Scale bars: 50 μm. **d** ROS levels as a function of time in larval CNS after injection with female Lb venom at a 1:50 dilution. The dashed line indicates the ROS levels of the 1×PBS-injected larvae, which was used as a control ($n = 3$ replicates, at least seven *Drosophila* CNSs were examined for each individual). Data are presented as mean values ± SD, and significance was analysed by one-way ANOVA followed by Šidák's multiple-comparison test (1 h: $P = 0.0002$; 2 h: $P = 4.8 \times 10^{-5}$; 3 h: $P = 0.0307$; 4 h: $P = 0.9998$; 5 h: $P = 0.8802$; 6 h: $P = 0.9999$; 12 h: $P = 0.9999$; 24 h: $P = 0.9949$; *$P < 0.05$; ***$P < 0.001$; ns: not significant). **e** Representative images of CNSs from Lb-parasitized larvae with or without expression of catalase from ubiquitously expressed *Actin-GAL4*, neuron-specific *Elav-GAL4* or haemocyte-specific *Hml-GAL4*. The CNS was stained with DCFH-DA. Scale bars: 50 μm. **f** Quantification of ROS levels in the CNSs of larvae with or without expression of *UAS-Catalase* using different *Actin-GAL4* for ubiquitous expression, *Elav-GAL4* for expression in neurons and *Hml-GAL4* for expression in haemocytes. Plotted is the mean intensity from fifty areas within each CNS ($n = 12$ *Drosophila* CNSs were examined for each individual). Data are presented as mean values ± SD. Significance was determined by two-sided unpaired Student's *t* test (WT: $P = 0.4986$; *Actin-GAL4*: $P = 2.0 \times 10^{-13}$; *Elav-GAL4*: $P = 1.0 \times 10^{-15}$; *Hml-GAL4*: $P = 0.8557$; ***$P < 0.001$; ns: not significant). **g** Escape indices for *Drosophila* hosts of the indicated genotypes. The number of larvae that exhibited escape behaviour was recorded from 30 min to 75 min in the presence of parasitoids. Left to right: $n = 288, 250, 368, 238, 235, 403, 448$ and 238. At least three biological replicates were performed. Data are presented as mean values ± SD. Significance was determined by two-sided unpaired Student's *t* test (WT: $P = 0.7217$; *Actin-GAL4*: $P = 0.0004$; *Elav-GAL4*: $P = 3.7 \times 10^{-5}$; *Hml-GAL4*: $P = 0.5762$; ***$P < 0.001$; ns: not significant). **h** Western blot analysis of EsGAP1 in the parasitoid venom apparatus and carcass. The representative images out of three independent replicates are displayed. CBB: Coomassie Brilliant Blue. **i** Representative images of EsGAP1 (red) immunolocalization in the nonparasitized larval brain (CK) and parasitized host brain (parasitized) 1 h after exposure to female Lb. Nuclei stained with DAPI (blue). Scale bars: 20 μm. **j** Characterised EsGAP1-staining spots in the CNS of parasitized and nonparasitized hosts ($n = 20$ biologically independent samples were examined for each group). Data are presented as mean values ± SD. Significance was determined by two-sided unpaired Student's *t* test ($P = 1.0 \times 10^{-15}$; ***$P < 0.001$). **k** Representative CNS image of *UAS-EsGAP1* or *Elav-GAL4 > UAS-EsGAP1* larva stained with DCFH-DA. Scale bar: 50 μm. **l** Quantification of ROS levels in the CNSs of *UAS-EsGAP1* or *Elav-GAL4 > UAS-EsGAP1* larvae. Each plot indicates the mean intensity from fifty areas within each CNS ($n = 10$ per phenotype). Data are presented as mean values ± SD. Significance was determined by two-sided unpaired Student's *t* test ($P = 1.0 \times 10^{-15}$; ***$P < 0.001$). **m** Representative images of CNS from larvae parasitized by *dsGFP*-treated or *dsEsGAP1*-treated female *Lb*. The CNS was stained with DCFH-DA. Scale bars: 50 μm. **n** Quantification of ROS levels in the CNSs of larvae that parasitized by *dsGFP*-treated or *dsEsGAP1*-treated female Lb. Each plot indicates the mean intensity from fifty areas within each CNS ($n = 10$ per group). Data are presented as mean values ± SD. Significance was determined by two-sided unpaired Student's *t* test ($P = 3.6 \times 10^{-13}$; ***$P < 0.001$). Source data are provided as a Source data file.

To determine whether wasp venom is responsible for ROS induction in parasitisation, we next injected female parasitoid venom extracts directly into 2nd instar *D. melanogaster* host larvae. A dose response was observed, and as little as a 1:50 dilution of the venom extract derived from a single female wasp induced ROS levels in the host CNS similar to those in parasitized *D. melanogaster* larvae (Fig. 3c and Supplementary Fig. 5). Moreover, the ROS levels were highest at 1 h post injection and declined over time, returning to the levels of control larvae after 3 h (Fig. 3d). Thus, our results not only demonstrate that parasitoid venom induces the production of ROS in the host neural system but also suggest that transient ROS induction might promote the short-term escape behaviour of parasitized hosts.

**Induction of ROS contributes to escape behaviour via EsGAP.** To further investigate whether ROS induction in the CNS is functionally important to escape behaviour, we sought to suppress ROS by overexpressing the antioxidant enzyme catalase. When *Actin-GAL4* was used to drive *UAS-Catalase* ubiquitously in Lb-parasitized larvae, elevated ROS production was significantly suppressed (Fig. 3e, f). In contrast, when we used *Hml-GAL4*, a haemocyte expression driver, ROS elevation was not suppressed at all (Fig. 3e, f), while dramatic suppression was seen when we used *Elav-GAL4*, a specific driver in neurons of the CNS (Fig. 3e, f). We then measured the escape performance of these larvae under our standard conditions from 30 to 75 min after exposing host larvae to female wasps. Strikingly, the escape index was reduced from above 30% to below 10% specifically for the two genotypes in which ROS induction was suppressed (Fig. 3g).

These results suggest that ROS induction is a key intermediate in provoking escape behaviour.

To detect whether the EsGAP proteins could directly trigger ROS accumulation in the host CNS, we generated an antibody against EsGAP1 that showed the strongest effect on reducing escape behaviour under knockdown conditions across the three *EsGAP* genes. Immunohistochemistry and western blot analyses support that EsGAP1 is expressed specifically in Lb venom glands (Fig. 3h and Supplementary Fig. 6). Antibody staining further showed massive spots of EsGAP1 in the brains of parasitized hosts, in comparison to the much weaker, non-specific signals in nonparasitized hosts (Fig. 3i, j). We then generated *UAS-EsGAP1* transgenic flies and introduced *Elav-GAL4* to specifically express EsGAP1 protein of Lb in *Drosophila* neurons. The results showed significantly higher ROS levels in the CNS of *Drosophila* larvae expressing both *Elav-GAL4* and *UAS-EsGAP1*, in comparison to the *UAS-EsGAP1* control (Fig. 3k, l). In addition, knockdown of EsGAP1 largely suppressed ROS induction in parasitized host larvae (Fig. 3m, n). These results strongly suggest that the EsGAP venom proteins are directly responsible for neuronal ROS induction upon parasitisation by Lb.

**EsGAP leads to superparasitism avoidance in multiple *Drosophila* species.** To return to the main focus of this study, we tested whether EsGAP directly confers superparasitism avoidance in Lb. We dissected 2nd instar larvae from *D. melanogaster* and counted the number of parasitoid eggs in each host after 60 min of infection with *dsGFP*-treated Lb ($P_{dsGFP}$) and *dsEsGAP1*-treated Lb ($P_{dsEsGAP1}$). Indeed, we found that there was a significant increase in superparasitism in *D. melanogaster* hosts parasitized with $P_{dsEsGAP1}$, and that *D. melanogaster* hosts treated with

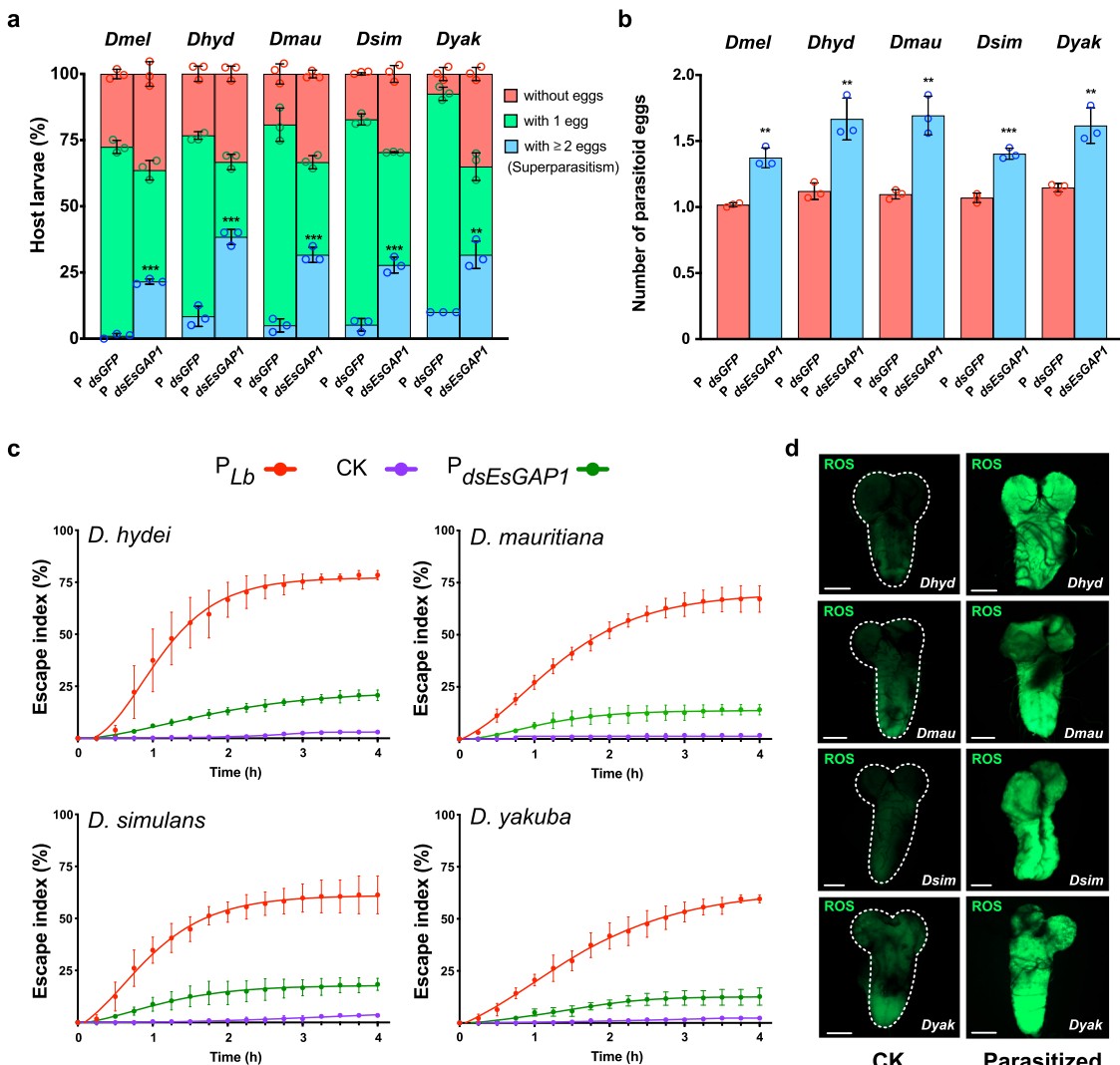

**Fig. 4 Lb utilises host escape behaviour to avoid superparasitism. a** Percentage of host larvae of five *Drosophila* species containing no, one or two or more wasp eggs after 60 min exposure to *dsGFP*-treated female wasps (P$_{dsGFP}$) and *dsEsGAP1*-treated Lb female wasps (P$_{dsEsGAP1}$). P$_{dsGFP}$ was used as a control. The presence of two or more wasp eggs in one host was considered superparasitism. Left to right: $n = 187, 201, 120, 120, 120, 120, 146, 122, 120$ and $120$. Three biological replicates were performed. Data are presented as mean values ± SD. Significance was analysed by two-sided unpaired Student's *t* test (P$_{EsGAP1}$ with *Dmel*: $P = 1.3 \times 10^{-5}$; P$_{EsGAP1}$ with *Dhyd*: $P = 0.0004$; P$_{EsGAP1}$ with *Dmau*: $P = 0.0003$; P$_{EsGAP1}$ with *Dsim*: $P = 0.0005$; P$_{EsGAP1}$ with *Dyak*: $P = 0.0020$; **$P < 0.01$; ***$P < 0.001$). **b** Average number of parasitoid eggs in control (red columns, P$_{dsGFP}$) and escape-depleted (blue columns, P$_{dsEsGAP1}$) *Drosophila* host larvae after 60 min in the presence of wasps. Three biological replicates were performed. Data are presented as mean values ± SD. Significance was analysed by two-sided unpaired Student's *t* test (P$_{EsGAP1}$ with *Dmel*: $P = 0.0013$; P$_{EsGAP1}$ with *Dhyd*: $P = 0.0052$; P$_{EsGAP1}$ with *Dmau*: $P = 0.0024$; P$_{EsGAP1}$ with *Dsim*: $P = 0.0005$; P$_{EsGAP1}$ with *Dyak*: $P = 0.0041$; **$P < 0.01$; ***$P < 0.001$). **c** The escape indices of hosts of four *Drosophila* species exposed to no wasps as a control (CK, purple curve, $n = 388$ for Dhyd, $n = 226$ for Dmau, $n = 196$ for Dsim, $n = 304$ for Dyak), to normal Lb female wasps for 4 h (P$_{Lb}$, red curve, $n = 296$ for Dhyd, $n = 286$ for Dmau, $n = 178$ for Dsim, $n = 312$ for Dyak) and to *dsEsGAP1*-treated female wasps (P$_{dsEsGAP1}$, green curve, $n = 300$ for Dhyd, $n = 280$ for Dmau, $n = 200$ for Dsim, $n = 300$ for Dyak). The number of escaped hosts was recorded and summed every 15 min. Three biological replicates were performed. Difference in escape index between each treatment was determined significance by one-way ANOVA along with Fisher's least significant difference tests. See detailed significance values of each time point in Supplementary Table 5. **d** Representative images of CNSs from nonparasitized control (CK) and Lb-parasitized larvae (parasitized) of different *Drosophila* species ($n = 10$ *Drosophila* CNSs were examined for each individual). The CNS was stained with DCFH-DA. Scale bars: 50 μm. Dmel *D. melanogaster*, Dhyd *D. hydei*, Dmau *D. mauritiana*, Dsim, *D. simulans*, Dyak *D. yakuba*. Source data are provided as a Source data file.

P$_{dsEsGAP1}$ had ~1.37 eggs, which is an increase of 34% in comparison to the P$_{dsGFP}$-treated control hosts (Fig. 4a, b). This result indicates that EsGAP1 effectively leads to superparasitism avoidance in Lb.

Although Lb is a representative specialist parasitoid that mainly parasitises *D. melanogaster*, we surprisingly found that its parasitisation can induce escape in hosts of other *Drosophila* species. We tested the escape preferences in four additional *Drosophila* species, including *D. hydei*, *D. mauritiana*,

*D. simulans* and *D. yakuba*, upon Lb parasitisation. All the four tested species showed a considerable escape index in the 4 h period following exposure to Lb, along with high accumulation of neuronal ROS in the parasitized hosts (Fig. 4c, d). Similar to *D. melanogaster*, knockdown of EsGAP1 in Lb adult females substantially decreased host escape behaviour (Fig. 4c). We further tested the role of EsGAP in superparasitism avoidance in these additional species by comparing the laid eggs between P$_{dsEsGAP1}$ and control hosts (P$_{dsGFP}$). As a result, we found that

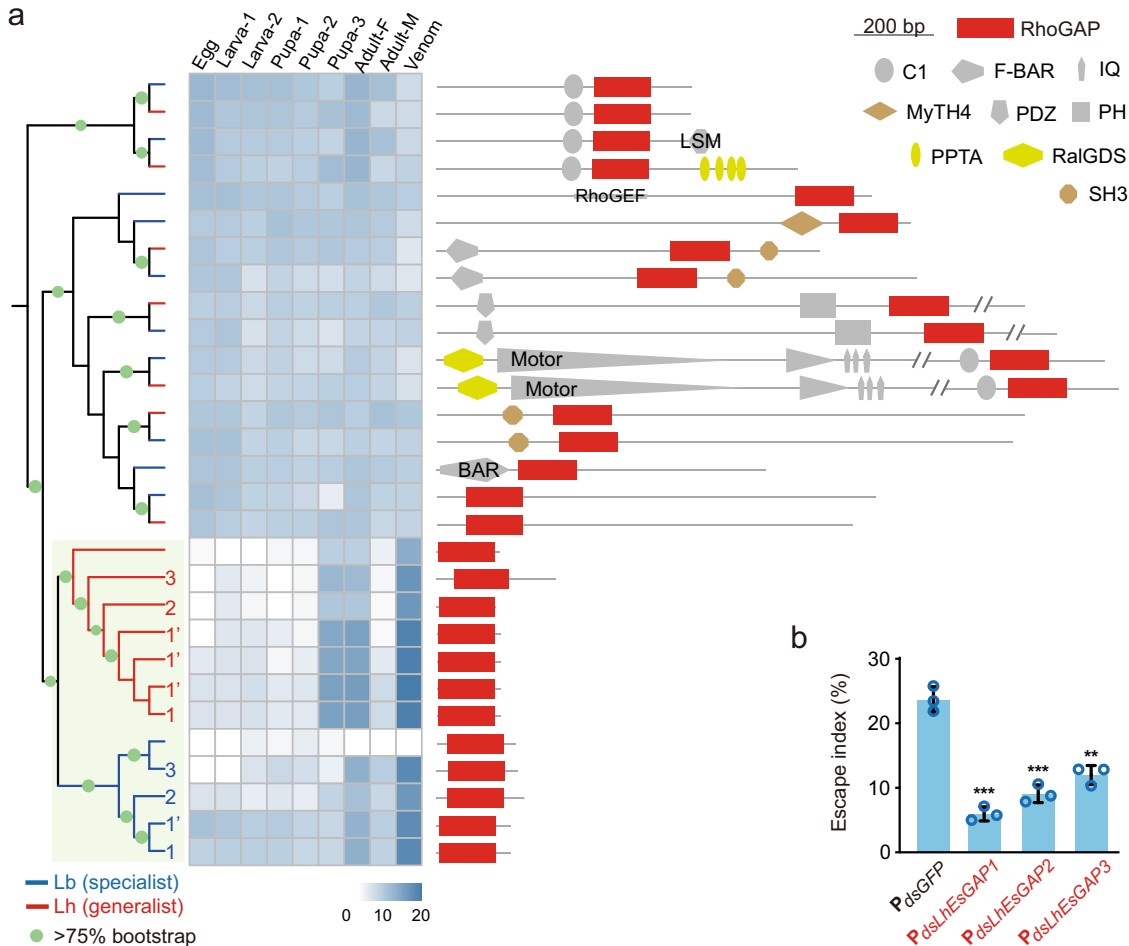

**Fig. 5 Molecular evolution of RhoGAP-containing genes in *Leptopilina*. a** Phylogenetic relationship across all potential RhoGAP domains encoded in Lh and Lb genomes. Branches being followed with a number indicate the corresponding termed *EsGAP* genes with functional evidence. Note that there are tandem duplicated copies of *EsGAP1* in both Lb and Lh, which share the near-identical sequence with *EsGAP1* and therefore are termed as *LbEsGAP1'* and *LhEsGAP1'*, respectively. The expression profile of each locus is shown in the heatmap. The actual gene length is indicated by the line and schematic gene architecture is shown with annotated domains. See detailed information in Supplementary Table 3. **b** Escape indices of host (*D. melanogaster*) that were parasitized by *dsRNA*-treated Lh. *LhEsGAP1*, LhOGS01638; *LhEsGAP2*, LhOGS20221; *LhEsGAP3*, LhOGS01640. *dsGFP*-treated Lb (P$_{dsGFP}$) was used as a control. The number of larvae exhibiting escape behaviour was recorded from 30 min to 75 min after exposure to Lh. Left to right: $n$ = 424, 357, 341 and 355 biologically independent host larvae. Three biological replicates were performed. Data are presented as mean values ± SD. Significance was determined by two-sided unpaired Student's $t$ test (P$_{dsLhEsGAP1}$: $P$ = 0.0002; P$_{dsLhEsGAP2}$: $P$ = 0.0004; P$_{dsLhEsGAP3}$: $P$ = 0.0011; **$P$ < 0.01; ***$P$ < 0.001). Source data are provided as a Source data file.

P$_{dsEsGAP1}$ leads to a significant increase in superparasitism in all tested host species (Fig. 4a), with on average ~1.67 eggs in *D. hydei* (increased 49% in comparison to control), ~1.69 eggs in *D. mauritiana* (increased 54% in comparison to control), ~1.40 eggs in *D. simulans* (increased 31% in comparison to control) and ~1.62 eggs in *D. yakuba* (increased 41% in comparison to control) (Fig. 4b). Collectively, knockdown of EsGAP attenuated the superparasitism avoidance of Lb in all investigated *Drosophila* species, suggesting that EsGAP-triggered host escape might be a general mechanism for successful parasitisation of *Drosophila* larvae (Fig. 4a).

**EsGAPs were newly recruited and evolved with novel functions.** We last investigated whether the host manipulation strategy has been utilised by an extended range of drosophilid parasitoids and analysed how these *EsGAP* genes evolved across parasitoid species (Fig. 5a and Supplementary Fig. 7). Although these genes were annotated with a RhoGAP domain (PF00620), their orthologues in related hymenopteran species were not found, even in the

closely related species *L. heterotoma* (Lh), a generalist *Drosophila* parasitoid. We observed a high level of divergence, particularly at the amino acid level, among the three *EsGAP* genes of Lb, indicative of a rapid evolutionary rate of this gene family. Given that the rapidly evolved genes might be underrepresented in the automatic annotation approach, we scanned this domain across the genomes of Lh and Lb. In total, 15 loci were found to encode this domain in the Lb genome, while 14 were found in the Lh genome (Supplementary Table 3). Phylogenetic analysis showed two distinct evolutionary clades for these *Leptopilina* RhoGAP domain-containing genes (Fig. 5a). Some of them were likely to be inherited from their common ancestor and evolved with the divergence between Lb and Lh, since there were one-versus-one orthologue pairs between the two species (Fig. 5a, branches without shadow). The other sublineages, including *EsGAP1-3* (Supplementary Fig. 7), of functional importance in Lb formed two lineage-specific expanded sublineages (Fig. 5a, shadowed branches). Interestingly, genes within the sublineage containing one-to-one orthologues were generally of much longer gene length and accompanied with other domains, such as C1, SH3,

BAR, PH and PDZ (Fig. 5a, non-shadowed), while the lineage-specific expanded genes encode only the RhoGAP domain and lack any other gene structures (Fig. 5a, green shadowed). Furthermore, in comparison to the widespread expression of the one-to-one orthologues, most members of the lineage-specific expanded genes presented evident specialised expression in venoms (Fig. 5a). Both gene architectures and expression patterns suggested distinguished functional roles between the two sublineages.

Public databases, such as GenBank and Pfam, document a widespread distribution of RhoGAP domains across eukaryotes, involving in fundamental regulation of cell differentiation and development[42–44], and their complete absence in prokaryotes. Correspondingly, we found that the majority of these genes were of long length, ranging from 500 to 1200 amino acids, approaching those in the orthologous sublineage of *Leptopilina*. Typical architectures of such genes also come with other functional domains, such as PH, FCH, BAR and SH2 (www.pfam.org). By contrast, genes that consist of RhoGAP domain only were limited to a few hymenopteran species based on our independent searches. We performed a comprehensive phylogenetic analysis across eukaryotes, involving all characterised hymenopteran candidates and those from representative species that fill all major evolutionary gaps. The phylogenetic topology strongly suggests multiple origins of these subgroups of *Leptopilina* RhoGAPs (Supplementary Fig. 7). Four genes within Lb–Lh orthologous subgroup were placed together with those of higher animals and most Hexapoda, forming a tightly clustered sublineage that represents the common Metazoan RhoGAP genes (Supplementary Fig. 7). By contrast, the other members of Lb–Lh orthologous sublineage and all members of the lineage-specific expanded subgroup, i.e., the venom-specific RhoGAP domain-containing genes were placed in scattered sublineages related to non-metazoan eukaryotes. These sublineages did not correspond to the evolution between species and were placed distantly from each other (Supplementary Fig. 7), leading us to propose two possible explanations: (1) these lineage-specific genes originated from lateral gene transfer (LGT) events via recruiting the RhoGAP domain from distant species such as some kinds of protists; (2) they arose from the duplication of the typical RhoGAP genes and experienced rapid, divergent evolution subsequently, but were inappropriately placed in the phylogenetic analysis due to the effect of long-branch attraction. We further performed the phylogenetic analysis by focusing on hymenopteran genes, which similarly formed a distinct, monophyletic clade for these lineage-specific RhoGAP genes (Supplementary Fig. 8). These unusual sublineages were also found in a few other parasitoid species (Supplementary Fig. 7). Thus, LGTs or duplications might have occurred multiple times and originated independently. Since the acquisition, these genes evolved following different trajectories that resulted in divergent fates. Some of them, i.e., *EsGAP1-3* underwent specialisation for expression in venom and ultimately gained novel functionality, i.e., manipulation of host behaviour in parasitoids. The preservation and duplication of these novel genes strongly suggest their selective advantageous roles during evolution.

**EsGAP homologs lead to superparasitism avoidance in the generalist *Leptopilina*.** The functionalized Lb-specific expanded sublineage was placed as a sister clade to another expanded sublineage of Lh. In this monophyletic group, most members showed specialised expression in venom, indicating a similar evolutionary scenario of these genes in Lh (Fig. 5a). To test this hypothesis, we further selected three RhoGAP genes from the Lh-specific expanded lineage with the highest expression in venom

(*LhOGS01638*, *LhOGS01640*, and *LhOGS20221*) to explore their associations with host escape. Similarly, we used the RNAi technique to knock down each of the three Lh RhoGAP genes individually. The qRT-PCR results showed that the expression levels of these RhoGAP genes were significantly reduced in emerging Lh adults upon RNAi, albeit with a similar non-specific reduction in expression across them (Supplementary Fig. 9). We next used dsRNA-treated Lh female wasps to parasitise *Drosophila* host larvae, with *dsGFP*-injected wasps as a control. As expected, significant suppression of the escape index in parasitized host larvae was found with knockdown of each of the three Lh RhoGAP genes (Fig. 5b). These genes are thus termed as *LhEsGAP1-3* according to the suppressing effect of RNAi (Fig. 5b). Our results suggest that the functionalization of RhoGAPs is strongly associated with the specialised expression in venoms and that these functionalized RhoGAPs have an ancestral origin prior to the divergence between Lb and Lh that show differences in host range. Independently massive duplication along each lineage strongly indicates that these neofunctionalized domains are selectively advantageous in both specialist and generalist *Leptopilina* parasitoids.

## Discussion

How parasitoids avoid superparasitism has long been pondered[10,12,21]. It has been commonly proposed that female parasitoids have evolved the ability of host discrimination to distinguish unparasitized hosts from parasitized hosts[19,45–47]. Host discrimination has been well-documented in ~200 wasp species[12]. This ability helps parasitoids detect the parasitized status of hosts and make a strategic decision to reject or accept previously parasitized hosts to maximise the rate of fitness gain. Previous studies hypothesised that host discrimination might be mediated by external or internal marking pheromones left by previous parasitoids because chemical cues were detected and recognised by the antennae or the ovipositor of the latter parasitoid[48–51]. However, whether these chemical cues are truly associated with host discrimination remains unclear. In this study, we surprisingly found that *Leptopilina* parasitoids avoid superparasitism by manipulating the short-term escape behaviour of hosts, a strategy not relying on detection of the parasitisation status of hosts. Since multiple parasitisation induces a higher death rate of hosts (Supplementary Fig. 1), both the "self" superparasitism and the conspecific superparasitism lead to a high risk for the survival of parasitoids. The evolved strategy of manipulating host escape could enable *Leptopilina* females to effectively reduce encounters of previously parasitized hosts and, consequently, minimise the risk of superparasitism. In addition, *Leptopilina* parasitoids are proovigenic and short-lived; the females only produce a limited load of eggs[52]. Rejecting parasitized hosts via manipulating them to escape would allow parasitoids to maximise the number of eggs that successfully develop into adults before death. Correspondingly, we note that the portion of hosts without wasp eggs, i.e. those nonparasitized flies, is also significantly elevated upon the exposure to *dsEsGAP*-treated Lb compared to the control group (Supplementary Fig. 10). This further supports that a superparasitism avoidance strategy enhances the efficiency of parasitoids.

It is common for parasites and parasitoids to manipulate host behaviours for their own advantages, such as locomotion, foraging, reproduction and social interactions[53–55]. Some of the induced behavioural responses are incredible, e.g. the "suicidal behaviour" that facilitates the movement of parasites to their next hosts[56–59] and the "bodyguard behaviour" that protects the host from its natural enemies[60,61]. On the other hand, some behavioural responses are only beneficial for the host but not

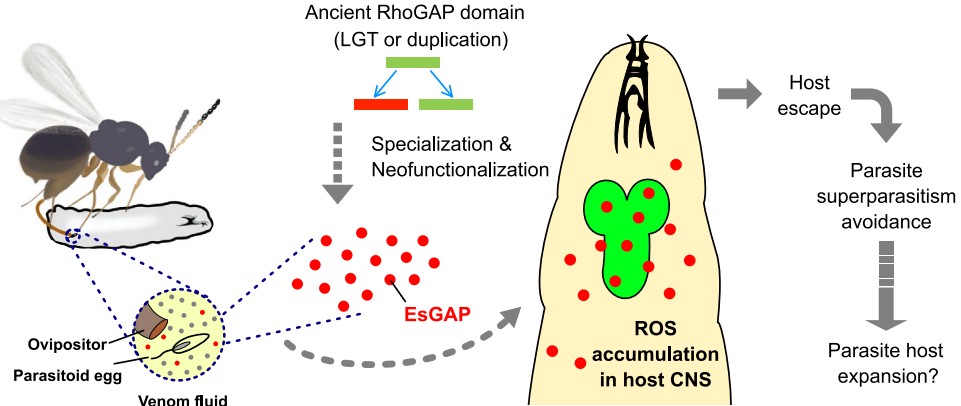

**Fig. 6 The proposed evolutionary scenario of superparasitism avoidance in *Leptopilina*.** EsGAP proteins are injected into the host along the parasitisation of *Leptopilina* wasps. The gene encoding this type of proteins was ancestrally originated, from either lateral gene transfer (LGT) from distantly related species or the duplication event of existed paralogs. Since the acquisition, this gene experienced further duplication and rapid evolution. Of duplicated copies, the ones (red) of expression specialisation in the venom were ultimately neofunctionalized. Injected EsGAP proteins induce the ROS accumulation in the host CNS that leads to the escape behaviour of hosts. Through manipulating the escape behaviour of parasitized hosts, parasitoids avoid superparasitism and intraspecific competition. As a long-term effect, intraspecific competition might lead to increased resource use diversity (see refs. [76-78] and our 'Discussion'). The images were crafted using Affinity Designer v1.8.6 and Procreate v4.2.2.

advantageous for the parasites, e.g. the parasitized bumblebee remains in the cold to hinder the development of the parasitoid eggs[62]. Despite broad interests, little is known about the proximal mechanisms underlying these fantastic manipulated behaviours. Here, our study introduces an exquisite example of manipulation of host behaviour by parasitoids, which is utilised as an adaptive strategy to avoid superparasitism. More importantly, we successfully characterise the genetic bases underlying how parasitoids manipulate hosts to escape. We find that parasitoids inject an unusual class of RhoGAP proteins to the host to induce the accumulation of ROS in the CNSs of hosts (Fig. 6). To defend against infection, ROS are generated in the host either within mitochondria or through the oxidative burst process mediated by the NADPH oxidase complex[63]. It was recently found that wasp infection increases ROS levels in host lymph glands, leading to lamellocyte differentiation and encapsulation of the immune response[64,65]. Correspondingly, we found that there is an increase in ROS in lymph glands approximately 12 h after infection (Supplementary Fig. 11). We also observed high accumulation of ROS levels in the host CNS, but not in other organs, soon after Lb infection (Fig. 3a, b and Supplementary Fig. 4), and that the accumulation of ROS is necessary to initiate the escape behaviour of parasitized hosts. It is still unknown how the host mediates ROS level to normal after being parasitisation. Interestingly, the escape behaviour we observed for parasitized host larvae in this study is very similar to that of anoxic larvae. It has been shown that hypoxia triggers ROS production in the *Drosophila* CNS[41]. Correspondingly, we note that one of the highly expressed VP genes encodes the superoxide dismutase (SOD) (Fig. 2a and Supplementary Table 2), a universal enzyme that controls the levels of a variety of ROS[66]. We thus propose a possible role of SOD in mediating ROS level in CNS, although we note the additional role of this VP gene in sabotaging the immune responses of hosts[67].

Our study suggests that the genes conferring host escape are novel venom genes that possibly originated from lateral gene transfer or the co-option of duplicated genes and all experienced subsequent specialised expression in venom and recent duplication. This evolutionary pattern highlights the importance of expression specialisation and functional switching in the evolution of novel gene function in venom[35]. More importantly, these signatures allow us to address the long-term debate regarding the

evolutionary context of superparasitism avoidance. Despite the multiple aspects by which it affects fitness, the selective advantage of superparasitism avoidance has long been challenged. Theoretically, superparasitism might be advantageous under some circumstances, such as the probability of elimination of strong intraspecific competitors or the enhanced suppression of the host immune response that ensures successful parasitism[12,68,69]. Moreover, an inheritable virus named Lb filamentous virus (LbFV) was discovered in some Lb strains that is responsible for an increase in superparasitism to enhance horizontal transmission[13,26,27]. These findings presented an alternative hypothesis that superparasitism is probably adaptive for symbiotic viruses but not parasitoids, i.e. the superparasitism avoidance in Lb is probably manipulated but adaptive. However, the Lb strain that showed clear superparasitism avoidance in this study was checked without infection with LbFV, ruling out the possibility of manipulation in our system. Instead, our functional and evolutionary studies showed that this behaviour was conferred by a small group of non-typical RhoGAP domain-containing genes that were recruited or co-opted by venom and evolved new functions. These genes share a similar gene expression pattern, i.e., specialised expressed in venom and are undergoing duplication in two *Leptopilina* species independently. Transfer of foreign DNA and gene duplication is not uncommon in eukaryotes, but the outcomes of the majority of these events are becoming silenced due to non-functionalization or functional redundancy. The retained genes generally showed signatures of acquisition of a beneficial function and preservation by natural selection. In addition to superparasitism avoidance, other adaptive traits or behaviours of parasitisation might co-drive the neofunctionalization of parasitoid genes. Interestingly, we note that one of the duplicated *EsGAPs*, *EsGAP2*, shows 99.7% sequence identity to a previously described Lb *RhoGAP* gene (GU300066.1), which was found as virulence factors to suppress the host immune response by affecting host hemocytes and preventing the encapsulation response[68,70,71]. Further studies might help address the possible diversified functions across these *EsGAP* genes. On the other hand, we note that the parasitoid strains used in this study have been maintained in the laboratory condition for a relatively long period[69,72,73]. Selection on competition-related traits, such as superparasitism avoidance, might be stronger under laboratory condition than in the wild. Although the presence in both Lb and

Lh strongly suggests an ancient origin of these host escape-related *EsGAP* genes, we cannot rule out the impacts of non-wild conditions on the subsequent evolution of these genes, e.g. the very recent duplication events on *LbEsGAP1* and *LhEsGAP1* (Fig. 5a).

Host shift or changed host range is one of the main ecological forces in driving speciation[74,75]. For *Leptopilina* parasitoids, a variety of host ranges have been documented for different species. Our functional evidence showed that this strategy allows the specialist Lb to manipulate the escape behaviour of other *Drosophila* hosts in addition to *D. melanogaster*, and that this strategy is also utilised by the generalist parasitoid Lh which is able to utilise a wide range of *Drosophila* hosts. These results, along with the monophyletic pattern of these lineage-specific expansions, strongly suggest that the common ancestor of the specialist and generalist evolved the host escape strategy to avoid superparasitism. Since the competition was proposed to increase resource use diversity within a natural population[76–78], this ancestrally evolved competition syndrome was likely to induce parasitoids to colonise additional hosts. Interestingly, although Lb is only able to parasitise *D. melanogaster* and its close relatives, it is not exceptional to find laid eggs of Lb in other *Drosophila* hosts, suggesting the potential of host range expansion in this specialist species. However, successful parasitisation of additional hosts relies on further adaptation, e.g. how to synchronise development with the host and how to combat with the host immune system. Correspondingly, a related study showed that the more powerful active immune suppression, which is absent in the specialist, allows broad parasitisation in the generalist Lh[34].

In conclusion, we have discovered a novel strategy by which venom protein emitted by the parasitic wasp induces behaviour in the host to limit the risk of superparasitism. The class of EsGAP venom proteins induces ROS production in the host, which leads the host to exhibit short-term escape behaviour. These findings allow us to propose a novel model of how parasitoid wasps implement superparasitism avoidance, and we strongly suggest that this strategy is selectively advantageous. These dependencies may also be utilised to increase the reproductive efficiency of agriculturally important parasitic wasp species and enhance the biological control of insect pests by parasitoids.

## Methods

**Insects**. The parasitic wasp strains *L. boulardi* (Lb) and *L. heterotoma* (Lh) were kindly provided by Dr. Dan Hultmark and Dr. István Andó, respectively. Both strains are maintained on *D. melanogaster* (W[1118] strain) as regular hosts under the conditions as described[73]. Briefly, ~100 mated *Drosophila* females were allowed to lay eggs within a plastic fly bottle (6-ounce square bottom) for 2 h and removed from the bottle. The fly bottles were then put at 25 °C, 50% humidity, and 16 h:8 h light:dark cycle (16 L:8 D) to allow egg hatching. 10 mated wasp females were introduced to the early 2nd instar *Drosophila* larvae for parasitisation for 4 h and removed from the bottle. The fly bottles were put back at 25 °C, 50% humidity, and 16 L:8 D to allow the development of parasitoids. The newly emerged wasps were collected into vials containing apple juice agar medium until exposure to hosts. The apple juice agar recipe: 27 g agar, 33 g brown sugar and 330 ml pure apple juice in 1000 ml of diluted water. Both strains were checked with Lb filamentous virus (LbFV) by mapping whole-genome sequencing reads with the virus reference. Genome sequencing reads of Lb is the sequencing data for polishing the assembly of long reads (see below), and the reads of Lh was generated by the previous study[34].

To generate the *UAS-EsGAP1* transgenic flies, the coding region of *EsGAP1* was amplified from cDNA of *L. boulardi* venom glands using the primers listed in Supplementary Data 1. The resulting cDNA fragment was inserted into the *pUAST-attB* vector digested with *NotI* and *XbaI*. The plasmid was injected into *Drosophila* embryos and specifically integrated into the *attP2* site (BL#8622). All *Drosophila* species, including *D. melanogaster*, *D. simulans*, *D. yakuba*, *D. hydei* and *D. mauritiana*, were reared on the same standard cornmeal/molasses/agar medium at 25 °C within 6-ounce square bottom plastic fly bottles[79].

**Superparasitism assays**. Mated 3-day-old Lb females were allowed to parasitise early 2nd instar *Drosophila* larvae at different time courses, including 30, 45, 60, 75, 90, 120, 180 and 240 min for superparasitism assays in Fig. 1a, b, and 60 min only for superparasitism assays in Fig. 4a. The number of female wasps in each test was

strictly based on the parasite/host ratios of ~1:10 or ~1:20. The exact numbers of hosts and wasps were also provided in source data. Once the wasps were removed, the host larvae were dissected under the microscope to count the number of parasitoid eggs. When one host had more than two parasitoid eggs, this was recognised as superparasitism.

**Escape behaviour assays**. Mated 3-day-old *Leptopilina* females were anaesthetised and released to early 2nd instar host larvae with a parasite/host ratio of ~1:10 in fly food bottles (see the exact numbers of hosts and wasps for each assay in source data). We note that there is an immediately avoidance behaviour of *Drosophila* larvae upon the release of wasps, which is similar to the focused escape behaviour of this study. However, this immediate escape behaviour only occurs before the anaesthetisation relief of wasps and lasts less than 2 min, while the focused escape behaviours of this study are all observed after released parasitoids being completely recovered from anaesthesia and ready for parasitisation. Specifically, when *Drosophila* larvae began leaving the food and crawling on the bottle walls, they were removed (not being allowed to return to the food) and recorded as host escape behaviour. Wasps were removed from the bottle after 4 h, and the numbers of escapees were counted at 15 min, 30 min, 45 min and so forth (Figs. 1d and 4c). In certain experiments (Figs. 1g, 2b, 3g and 5b), the escapees were only recorded at the slot from 30 to 75 min after cohabitation with female wasps or injection with venom at different dilutions, which is the stage in which most of the parasitized host larvae presented escape behaviour. The escape index was calculated as the percentage of the escapees to the total released hosts.

**High-quality Lb reference genome**. To generate a high-quality reference genome, the genome of Lb was fully sequenced using long-read sequencing technology (PacBio). Lb has a tiny body size, so DNA was extracted from a pool of ~2000 male Lb adults to meet the requirements for library construction. Genomic DNA was prepared using a DNeasy Blood and Tissue Kit (Qiagen). The initial survey was analysed based on K-mer analysis of 9-Gb Illumina sequencing data (150 bp paired-end data). Jellyfish v2.2.3[80] was utilised in K-mer analysis with k = 17. A 20-Kb genomic library was constructed and sequenced by Berry Genomics on the PacBio Sequel platform following the manufacturer's protocol. A total of 59 Gb of raw reads were generated. The raw reads were initially subjected to error correction using Canu v2.0[81]. Following a previously optimised approach, we selected a subset of the ~40 X longest corrected reads for genome assembly using Flye v2.7-b1585[82] with 2 polishing iterations and other default parameters. Assembled contigs were further polished using pilon v1.2.3[83] with the default parameters. The completeness of the assembly was evaluated using both CEGMA v2.4[84] and BUSCO v3[85]. For BUSCO, both sets of Insect_odb9 and Hymenoptera_odb9 were used.

**Genome annotation**. Repeat contents were annotated using RepeatMasker v4.0.5 (http://www.repeatmasker.org) against Repbase v20140131[86] (Arthropod set) and the custom library generated by RepeatModeler v1.0.7 (http://www.repeatmasker.org). Protein-coding genes were predicted using the Maker pipeline[87]. Both full-length transcripts and high-coverage RNAseq data across all developmental stages were downloaded from GenBank (PRJNA624743). The full-length transcript evidence was generated by mapping to the reference genome using Minimap2 v2.1[88] with the parameters "-ax splice -uf --secondary = no -C5" and sorting exons by the module "collapse_isoforms_by_sam" of cDNA_Cupcake v6.4 (https://github.com/Magdoll/cDNA_Cupcake). The high-coverage RNAseq evidence was generated by mapping paired-end Illumina reads to the reference genome using HISAT2 v2.1.0[89] with the parameter "--dta" and sorting exons using StringTie v2.0[90] with the default parameters. Homologue evidence was obtained as described previously[34], including UniProt database and gene sets of *Drosophila melanogaster* (http://flybase.org/) and five hymenopteran species (*Apis mellifera*, *Fopius arisanus*, *Nasonia vitripennis*, *Polistes dominula*, and *Trichogramma pretiosum*)[91–95]. Ab initio predictors, AUGUSTUS v3.3.2[96] and SNAP v2006-07-28[97], were provided for Maker v2.31.10 to integrate the above evidence to generate an automated set of protein-coding genes for Lb. The wasp genomes generally encode a considerable proportion of novel genes, which was possibly missed by the Maker pipeline due to lacking of consistent evidence. We added genes that were predicted by AUGUSTUS and further supported by both full-length transcripts and assembled transcripts to the final gene set.

Protein-coding genes were annotated using BLASTP to search the best homologs against several public databases, including NCBI RefSeq, UniProt, and FlyBase[98]. Local InterProScan v5.13[99] was performed with all involved databases to search functional domains, Gene Ontology terms, and KEGG Orthology (KO) terms for each gene. Previously published transcriptome data[34] were used to profile the expression pattern of each protein-coding gene. These data covers most representative developmental stages of parasitoid wasps, including Egg, Larva-1 (days 1–3 larvae), Larva-2 (days 4–6 larvae), Larva-3 (days 7–9 larvae), Pupa-1 (days 1–3 pupae), Pupa-2 (days 4–7 pupae), Pupa-3 (days 8–10 pupae), Adult-F (day 3 adult females) and Adult-M (day 3 adult males). Transcriptome data of venoms was generated independently[34], in which the venom glands of 3-day-old adult female wasps were independently dissected in Ringer's saline solution on an ice plate under a stereoscope (Nikon). To quantify the expression profiles, Salmon v0.12.0[100] was used to determine the expression level as Transcripts Per Million

(TMP) for each sample. Transcriptome data of eight development stages of Lh was also used as parallel comparison, including Egg, Larva-1 (days 1–3 larvae), Larva-2 (days 4–9 larvae), Pupa-1 (days 1–3 pupae), Pupa-2 (days 4–7 pupae), Pupa-3 (days 8–10 pupae), Adult-F (day 3 adult females) and Adult-M (day 3 adult males).

**Identification of Lb venom proteins**. The venom reservoirs of 3-day-old Lb female wasps were dissected in Ringer's saline solution on an ice plate under a stereoscope (Nikon). The venom reservoirs were washed at least three times in Ringer's buffer and then pierced by fine forceps in a cell culture dish to enable the collection of crude venom fluid into an Eppendorf tube. After centrifugation at $3000 \times g$ at 4 °C for 1 min, the supernatant (venom fluid) was stored at −80 °C until use.

For LC-MS/MS experiments, venom fluid including both extracellular vesicle packed and nonpacked venom proteins from 300 Lb 3-day-old females was dissolved in 100 μl SDT lysis buffer (4% SDS, 100 mM Tris-HCl, 1 mM DTT, pH 7.6). After boiling for 15 min, the sample was centrifuged at $13,000 \times g$ at 4 °C for 40 min. The amount of protein in the supernatant containing venom proteins was quantified with the BCA Protein Assay Kit (Invitrogen); the concentration was 2.2 μg/μl. In total, 60 μl of supernatant was used in this experiment. The detergent and DTT were removed by repeated ultrafiltration (Microcon units) using UA buffer (8 M urea, 150 mM Tris-HCl, pH 8.0), 100 μl iodoacetamide (100 mM) was added to block reduced cysteine residues, and the samples were incubated for 30 min in the dark. The filters were washed three times in 100 μl UA buffer and twice in 100 μl 25 mM $NH_4HCO_3$ buffer. Finally, the protein suspensions were digested with 3 μg trypsin (Promega) in 40 μl 100 mM $NH_4HCO_3$ buffer overnight at 37 °C. The resulting peptides were desalted on C18 cartridges (Empore SPE Cartridges C18 (standard density), bed I.D. 7 mm, volume 3 ml, Sigma), concentrated by vacuum centrifugation, and reconstituted in 40 μl of 0.1% (v/v) formic acid.

LC-MS/MS analysis was performed on a Q Exactive mass spectrometer (Thermo Fisher Scientific) coupled to an Easy nLC (Thermo Fisher Scientific). A 6-μl aliquot of the peptide mixture was loaded onto a reverse-phase trap column (Thermo Fisher Scientific Acclaim PepMap100, 100 μm × 2 cm, nanoViper C18) connected to the C18-reversed-phase analytical column (Thermo Fisher Scientific Easy Column, 10-cm long, 75-μm inner diameter, 3 μm resin) in buffer A (0.1% formic acid) and separated with a linear gradient of buffer B (84% acetonitrile and 0.1% formic acid) at a flow rate of 300 nl/min. The eluted peptides were ionised, and the full MS spectrum (from $m/z$ 300–1800) was acquired by precursor ion scan using the Orbitrap analyser with a resolution of $r = 70,000$ at m/z 200, followed by 20 MS/MS scans with a resolution of $r = 17,500$ at $m/z$ 200. The MS raw files were translated into mgf files and searched against the transcriptome of the Lb venom gland using Mascot 2.2. MS/MS tolerance was set at 20 ppm, and trypsin was defined as the cleavage enzyme allowing no more than two missed cleavages. Carbamidomethylation of cysteine was specified as a fixed modification, and oxidation of methionine was specified as a variable modification.

Venom-protein genes were finally defined based on both transcriptomic[34] and proteomic evidence as described above. Genes with an expression of 7.3 TPM in the venom (N99 across venom transcriptome data) were defined as venom gland-expressed genes. VG-expressed genes that were able to be fully aligned to at least three proteomic peptides were defined as venom proteins (Supplementary Table 2).

**Analysis of the origin of EsGAP**. The protein sequence of EsGAP1 was used to search for potential homologs across all organisms. We performed BLASTP searches for potential homologs against the NCBI NR database. The BLASTP searches were performed in June 2020, which yielded 11,891 hits (under an $e$ value of 1e-5). All these hits were eukaryotic. To balance the sequence representativeness and quality of multiple alignments, we further performed BLASTP searches against the subset database by limiting the class of organisms. Thus, BLASTP searches were independently performed against the organisms of Parasitoida, Hymenoptera (excluding Parasitoida), Hexapoda (excluding Hymenoptera), Deuterostomia and Metazoa (excluding Deuterostomia or Hexapoda). All identified homologs were sorted based on the blast $e$ value. All Hymenoptera hits and hits of other groups of the lowest $e$ values were retained in the multiple alignment. The web aligner MultAlin[101] was used to perform multiple alignment. Visual alignment errors were manually removed, and multiple alignment was performed iteratively. The relatively conserved blocks of approximately 170 aa of each sequence were used to infer the ML phylogenetic tree using RAxML v8.2.10[102] under the JTT model. Another phylogenetic analysis involving all hymenopteran hits (127) from 70 species was independently performed using the same approach, in which 165 aa of well-aligned segments were retained for inferring the phylogeny.

The EsGAP genes were highly diverged from each other. To overcome the potential underrepresentation of related homologs in the genome, an independent TBLASTN-GeneWise iterative approach was used to annotate all potential loci encoding a homologue of the RhoGAP domain. Briefly, the protein sequences of EsGAP1-3 were used as seeds to perform TBLASTN searches against the Lb and Lh genomes. The positive loci ($e$ value < 1e-5) were extended by 1 kb and were subject to exon–intron architecture prediction using GeneWise v2.2.26[103]. These processes were performed iteratively for three rounds. Independent phylogenetic analysis of Lh and Lb homologs was performed as described above. Lh genome and transcriptome data were retrieved from the previous study[34].

**Double-stranded RNA preparation and microinjection**. Double-stranded RNA (dsRNA) was synthesised using the T7 RiboMAX Express RNAi System Kit (Promega) according to the manufacturer's instructions. The primers used, which are complementary to the Lb venom genes (*LB2_330_135, LB2_255_017, LB2_151_037, LB2_216_018, LB2_151_039* and *LB2_352_022*), three Lh venom genes (*LhOGS01638, LhOGS01640* and *LhOGS20221*) and *GFP* (control), are listed in Supplementary Data 1. The reaction mixture was incubated at 37 °C for 4 h and heated at 70 °C for 10 min, and double strands were allowed to anneal at room temperature for 20 min. Subsequently, dsRNA was treated with RNase and DNase I to remove the templates and purified with isopropanol. The purified dsRNA was quantified with a NanoDrop 2000 (Thermo Fisher Scientific). Approximately 20 nl of dsRNA (5 μg/μL) was injected into each fifth instar wasp larva using the Eppendorf FemtoJet 4i device with the following parameters: injection pressure = 900 hPa; injection time = 0.15 s. More than 100 Lb and Lh larvae were injected for each gene, and experiments were repeated independently three times. After the dsRNA-treated parasitoids emerged from the hosts, we used some female wasps in each case to test the host escape response and the others to verify target mRNA knockdown efficiency using qRT-PCR. The number of female wasps in each test was strictly based on the parasite/host ratios of ~1:10. The exact numbers of hosts and wasps are shown in the source data.

**Quantitative RT-PCR**. The total RNA was isolated from 15 venom glands of dsRNA-treated wasps using the RNeasy Mini Kit (Qiagen) according to the manufacturer's protocol. Quantitative RT-PCR was performed in the AriaMx real-time PCR system (Agilent Technologies) with the One Step qPCR SYBR Green Kit (Vazyme). Primers for amplifying 100–300 bp of each PCR product are listed in Supplementary Data 1. Reactions of the cDNA synthesis and the qPCR were carried out for 30 min at 50 °C, followed by 5 min at 95 °C, followed by 40 cycles of one-step PCR for 10 s at 95 °C and 30 s at 60 °C. The RNA levels of target genes were normalised to *tubulin* mRNA, and their relative concentrations were determined using the $2^{-\Delta\Delta Ct}$ method[104].

**In vivo ROS analysis**. Intracellular ROS levels were determined by measuring the oxidative conversion of cell permeable 2',7'-dichlorofluorescein diacetate (DCFH-DA) to fluorescent dichlorofluorescein (DCF) using the Reactive Oxygen Species Assay Kit (Beyotime Biotechnology). The tissues including CNS, epidermis, fat body, gut, hemocyte, lymph gland, malpighian tubule, salivary gland and trachea tube were dissected in 1×PBS from nonparasitized host larvae and the parasitized hosts post 1 h infection, respectively. Then the tissues were incubated in DCFH-DA probe at a final concentration of 10 μM for 15 min in the dark, followed by rinsing with 1×PBS three times at room temperature. Samples were mounted, and fluorescence images were immediately captured on a confocal microscope (LSM 800, Carl Zeiss). ImageJ software (National Institutes of Health) was used to determine the mean intensity of 50 areas within each CNS.

The ROS quantification in Fig. 3 was carried out as described in Zhang et al.[105] with some modifications. Briefly, the CNSs of seven larvae for each group were dissected and homogenised in 100 μL NP40 lysis buffer (GenStar) and centrifuged at $12,000 \times g$ for 10 min at 4 °C. A 30-μL aliquot of supernatant was incubated with 100 μL DCFH-DA solution (100 μM, diluted in 1×PBS) at 37 °C for 1 h. Fluorescence intensity was detected at a 488-nm excitation wavelength and 528 nm emission wavelength with the SpectraMax iD5 Multi-Mode Microplate Reader (Molecular Devices). The ROS levels were normalised to the amount of total protein in each sample, which was determined using a BCA Protein Assay Kit (Invitrogen).

**Western blot analysis**. Anti-EsGAP1 antibody was generated in rabbits against the peptide (KKNIIKNVLKSNKNKEKADL) derived from the Lb EsGAP1 protein. Sixty female Lb were dissected on an ice plate. Total proteins from the wasp venom apparatus and carcass were isolated in protein extraction buffer (Sangon Biotech). After centrifugation at $13,000 \times g$ at 4 °C for 10 min, the supernatant was loaded for SDS-PAGE analysis, and the proteins were transferred to PVDF membranes (Millipore). Membranes were incubated in a blocking solution (Tris-buffered saline containing 0.1% Tween 20, 2% BSA) for 3 h and probed overnight at 4 °C with anti-EsGAP1 primary antibody (1:500 dilution). Horseradish peroxidase-conjugated anti-rabbit IgG secondary antibody (Solarbio) was used at a dilution of 1:2000.

**Immunohistochemistry**. Venom glands of Lb and brains of *D. melanogaster* host larvae were dissected in PBS and fixed in 4% paraformaldehyde in PBS for 30 min, rinsed with PBST (PBS containing 0.1% Triton X-100 and 0.05% Tween 20), blocked with 1% bovine serum albumin in PBST, and stained overnight at 4 °C with anti-EsGAP1 primary antibody (1:500). Venom glands or brains were then washed three times in PBST and incubated with Alexa Fluor 594 secondary antibody (1:1000; Molecular Probes) for 2 h at room temperature. Samples were mounted in ProLong Gold Antifade Mountant with DAPI (Invitrogen). Fluorescence images were captured on a Zeiss LSM 800 confocal microscope and were processed using ImageJ and Photoshop (Adobe).

**Data analysis and statistics**. Statistical analyses were performed in GraphPad Prism version 7.0a (GraphPad Software), SPSS 26.0 Statistics and EXCEL. Data were analysed for statistical significance using one-way analysis of variance (ANOVA) along with Fisher's least significant difference test (Figs. 1d, 4c and Supplementary Fig. 2a), one-way ANOVA followed by Šidák's multiple-comparison test (Fig. 3b, d) and unpaired two-tailed Student's $t$ test in other experiments. Error bars indicate the standard deviation (SD), and the data sets are represented as the mean ± SD. Significance values are indicated as follows: $*P < 0.05$, $**P < 0.01$ and $***P < 0.001$.

**Ethics statement**. Only insects of *Drosophila* and *Leptopilina* parasitoid wasps were used in this study which are not subject to ethical approval.

**Reporting summary**. Further information on research design is available in the Nature Research Reporting Summary linked to this article.

## Data availability

The transcript sequences of three *Lb EsGAP* genes used in this study have been deposited in GenBank with the accession numbers MZ673645 (*EsGAP1*), MZ673646 (*EsGAP2*) and MZ673647 (*EsGAP3*). The genome assembly generated in this study has been deposited in GenBank BioProject under accession number PRJNA671782. The proteome data of the Lb venom fluids have been deposited in PeptideAtlas under the accession number PASS01481. Source data are provided with this paper.

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

## Acknowledgements

We thank Drs. Yves Carton, Marylène Poirié and Jean-Luc Gatti for communicating the biological and ecological characteristics of *Leptopilina* strains. We thank Drs. Dan Hultmark, István Andó and Qi Zhou for providing parasitoid lines or *Drosophila* species. We also thank Drs. Dan Hultmark, Daniel Kalderon, Marcel Dicke and Shusheng Liu for critical comments on the manuscript. This study was supported by the National Key R&D Program of China (2017YFD0200400), the National Science Foundation of China (Grants # 31622048, 31772522, 31630060 and 31672370), the Chinese Academy of Sciences (Grants # QYZDB-SSW-SMC029 and XDB27040205), and the Zhejiang Provincial Natural Science Foundation of China (LR18C140001).

## Author contributions

J.H. conceived the project. J.H., X.C. and Sh.Z directed and supervised the project. J.H. and Sh.Z. designed the studies. J.C., L.P. and Y.S. performed and analysed the escape assay and ROS experiments. Q.Z., Yu.Z., Si.Z. and Y.L. performed RNAi and qRT-PCR experiments. Z.L. and M.S. performed the immunohistochemistry and western blot experiments. Sh.Z., G.F., Yi.Z. and G.L. performed genomic and evolutionary analyses. J.H., J.C. and Sh.Z. interpreted the data and wrote the manuscript. All authors have read and approved the manuscript submission.

## Competing interests

The authors declare no competing interests.
