## [Peer Review File · Nature Communications]

Neofunctionalization of an ancient domain allows parasites to avoid intraspecific competition by manipulating host behaviourREVIEWER COMMENTS

Reviewer #1 (Remarks to the Author):

The Authors present a long read assembly of the Hymenopteran parasitoid of *Drosophila*, *Leptopilina boulardi*. The assembly itself is a great improvement from a currently available one on NCBI: scaffold n50 of 458kb vs the authors new assembly with a contig n50 of 3.6 Mb. Furthermore, that assembly appears to be very complete by both CEGMA and BUSCO. Unfortunately the authors do not provide (that I could find) which taxonomic level the analyses used (insecta or hymenoptera?). This can have a dramatic effect on assembly “completeness”, especially in *Leptopilina*. This is a minor concern, and does not detract from the paper’s conclusions or impact.

In parasitization assays, the authors observed host larvae leaving food and found that the majority of larvae were parasitized. Furthermore the fly larvae that didn’t leave the food suffered high mortality, presumably due to superparasitization. They hypothesize that the host leaving food could be superparasitization avoidance. Injected wasp venom elicit the same response, while mock injections do not. This indicates that the wasps are behaviorally manipulating their hosts. From the wasp assembly and venom annotation, there were three highly expressed and venom specific gene products containing RhoGap domains, thus the authors hypothesized these venoms could be responsible for the host manipulation. RNAi knockdown of the RhoGap containing genes confirms this hypothesis. It is surprising that all three knockdowns have a greater than 50% effect on their own. This could be due to threshold effects or perhaps non-specific knockdown of the other paralogues, which is not tested in the diagnostic RT-PCRs (Supp. Fig 2a and 6a). If the alternative knockdown controls were not done, this should be mentioned.

The authors show that parasitization, and venom specifically, induces ROS in the larvae CNS. ROS is directly involved in escape behavior as shown by flies genetically manipulated to suppress ROS also suppress escape behavior in the presence of wasps. RhoGap is directly implicated in CNS ROS accumulation by immunohistochemistry showing CNS localization and RhoGap knockdown in wasps preventing ROS accumulation. The authors attempt to address the ecological implications of wasp induced host “escape” behavior by assessing the number of superparasitized fly larvae when wasps have RhoGap knocked down. Reduction in RhoGap leads to reduced “escape” and increased number of eggs per larvae. They go further to characterize this effect in other *Drosophila* species, showing that the wasps are manipulating a conserved mechanism of fly behavior (possible hypoxia avoidance).

The phylogenetic analysis and gene family reconstruction of RhoGap venoms is perhaps the weakest and or least clear section. I have several concerns: (1) The blastP search strategy for RhoGap homologues resulted in very hymenoptera/hexopoda weighted sampling. This most likely leads to problems with long branch attraction, inflating the evidence of lateral gene transfer, especially for fast evolving genes like the venoms are proposed to be. If the tree topology is to be believed, the *Leptopilina* RhoGaps originated from 6 independent LGTs. Likewise, there are several other long branched hymenopteran and hexapod RhoGaps placed with the non-metazoans. It is much more likely that these long branched genes arose from duplications within hexapod or hymenopteran evolution, evolved rapidly, and are erroneously placed deep in the tree because of it. A search strategy including the conserved *Leptopilina* RhoGap genes would fill out the tree more evenly because it wouldn’t rely on identity to a fast evolving

gene. Perhaps an even better approach would be to pull available RhoGap sequences from the InterProScan database, subsample and attempt to align them with the Leptopilina RhoGap sequences. (2) In Figure 5a, if the Lh RhoGap sequences were also included (in a more evenly sampled tree), it would help to identify when the LGT or duplication happened. Leaving these out doesn't make sense to me and leads to unnecessary reader questions. (3) Using *Microplitis demolitor* (misspelled on line 495) as the outgroup explicitly makes Fig 5b contradict the placement of the RhoGap gene family expansions in 5a. 5a shows them outside of hymenoptera (*) while 5b shows them within hymenopteran RhoGaps. If the authors wish to maintain that the venom RhoGaps could have come from a LGT, this tree should be unrooted so that it doesn't start with the assumption that all Leptopilina RhoGaps could coalesce as expected by the species tree (no LGT required). Without branch lengths, node support, and more outgroup sequences, Figure 5b supports duplication leading to the RhoGap venoms without having to hypothesize the less likely and poorly supported LGT from a distantly related eukaryote. I recommend removing figure 5a because it poorly supports LGT due to methodological issues. I also recommend that 5b should include several other hymenoptera RhoGaps (evenly sampled) and use a well annotated insect (like fly) as an outgroup. This new tree with node support would be far superior to 5a. If the Lb and Lh species specific expansions do not cluster with the rest of the Leptopilina RhoGaps (or hymenoptera), this would be much better evidence of LGT, and adding more diverse sequences would be warranted. Without these changes, proposing LGT as a possible origin of the venoms is completely unnecessary, when duplication and neofunctionalization are a likely and supported hypothesis.

Finally, the authors mention that the species specific RhoGap domains are accompanied by several other domains, but the conserved RhoGaps are not. The identity of the extra domains for Lb and Lh would be interesting to mention in this paper. Since Lb and Lh RhoGap venoms come from a single Leptopilina ancestor and seem to induce ROS and escape behavior, do they have the same domain architecture? Do all of the venom RhoGaps have the same extra domain? Have some of the duplicated venom genes gained different domains?

I believe the work presented here is given sufficient detail to be reproduced. As far as I can tell the statistical tests are appropriate. Fig 3b and 3d could use a mixed model to gain more power, however the t-test is conservative, and the point is made. There are no stats for the time series larvae escape figures (1d and 4c). These figures are convincing by themselves, but stats would be nice.

In general, the authors work presented here is novel and of great benefit to the research community. They tell a full genotype to phenotype story WITH evidence of implications on host/parasite selective forces, which is sorely lacking in the field. I believe this paper is very strong and recommend publication if either (A) the LGT hypothesis and Figure 5a are deemphasized/removed or (B) a more robust analysis is added which supports LGT.

Reviewer #2 (Remarks to the Author):

Summary

This paper elegantly addresses the ecological, genetic, and physiological basis of an extended phenotype; namely, it shows that evolutionarily derived venom proteins in a parasitoid wasp induce a

behavioral shift in its host--*Drosophila* larvae. This shift is facilitated in part through venom-related induction of ROS in the host nervous system. The ecological function of the manipulation is that it lowers levels of superparasitism under lab conditions, likely leading to decreased competition for developing wasp larvae. The compelling mechanistic story explaining manipulation is supported by a diverse set of analyses, including broad infection and behavioral time courses, multiple “omics” approaches, genetic manipulation, histology, and phylogenetics. As it stands, most of the results in this paper are compelling and original, and all the statistical, phylogenetic, and molecular methods are appropriate. I commend the authors for offering such a well written and thorough manuscript. I believe the results will prove interesting to a wide readership. However, there are a few areas where additional description of results/methods would be helpful. I also suggest places where it seems prudent to adopt more cautious/nuanced interpretations of the ecological and fitness results. Finally, note a few irregularities/omissions in data that need to be explained.

Main comments:

1. -A main area of concern is that the authors provide little quantification of the negative fitness effects of superparasitism for the wasp. The issue of wasp fitness only seems to be addressed when considering those larvae that did not exhibit any escape behavior through the full course of each experiment. The authors speculate that the 74% mortality column in Fig 1f results from super parasitized hosts. The actual distribution of eggs per host per behavioral category (escape vs not escape) is not reported. Moreover, the number of wasps per host is not described for those wasps that do hatch (Fig 1e,f). If multiple wasps come out of larvae that are expressing the escape behavior, then it is only removing the selection from “excessive” superparasitism. If previous studies more fully quantified the fitness impact of competition via superparasitism in this system, then it needs it should be included in the discussion.

-There is also a discordance between Fig1a and 4a. The original result shows a small but non-zero level of superparasitism at 60min, but this appears to vanish when the authors begin contrasting host behavior when exposed to control vs EsGAP1-knockdown wasps. It also seems that the interpretation of superparasitism levels is somewhat contingent on the exposure time to parasitoids. For example, in Fig1a almost all of the hosts experiencing superparasitism at 4hr. Perhaps the disconnect with fig 4a is the ratio of flies to wasps? The methods suggest that 50 dsRNA-injected wasps were used for experiments, but the ratio of hosts-wasps is not explained.

2. Another point that could benefit from added ecological context concerns the origins of the long-term lab origins/propagation of the wasp strains. The *L. boulardi* (Lb) wasp strain appears to have been isolated in 1978, and the strain of the other species has been maintained in the lab since 2002. If the genome sequences came from this lab strain, was any attempt made to confirm that the EsGAP paralogs exist in wild caught wasps? And if so, are there same number of copies between lab and wild? Some of the earliest studies of large effect behavioral adaptation genes were found to be either only observed in the lab (e.g., rover/sitter polymorphism in *Drosophila*), or were actually driven by lab-induced selection (e.g., *npr1* induced clumping during foraging in *C. elegans*). While the presence of venom-specific EsGAP genes in both the generalist and specialist parasitoids suggest their common ancestor must have had at least one copy, it isn't hard to envision how superparasitism selection could be much stronger under lab conditions than in the wild, and that some of the molecular evolution reported here may be lab derived.

This is particularly important given that the authors repeatedly suggest the mechanisms revealed in this study are relevant to naturally occurring superparasitism.

3. There is a worrisome irregularity about the number of RhoGAP domain transcripts/proteins that are highly expressed in Lb venom. Only 3 loci are dissected in the knockdown experiments and reported as being highly expressed. But Fig5b show 4 highly expressed paralogs, and SupTable 3 shows a 4th gene (LB2_216_152) with the domain and the same high length ratio. It seems unlikely that this 4th locus would have been excluded from preliminary mechanistic experiments, but if so why? Even if preliminary results no effect of LB2_216_152 on ROS or behavior, those data should still be reported as they speak to the varying effects of that protein domain.

4. Lines 161-171. The RNAi knockdown data, including the qRT-PCR and behavioral measurements, are interesting and integral to the paper. However, there is a worry that the dsRNA oligos for each locus might not have had a specific effect. The authors only report data for individual dsRNA-EsGap pairs. It is important to know whether the extra reduction in escape behavior from knockdown of LB2_255_017 is due to the phenotypic effect of only that locus, or whether the LB2_255_017-dsRNA has a spillover effect that also reduces expression of the other two EsGap genes. The specificity of RNAi knockdowns varies greatly among species, and spillovers have been observed even with substantial sequence differences between loci. This speaks to the genetic architecture of the induced host phenotype (i.e., additive vs redundancy vs big+small effect loci).

5. Lines 186-188: the authors suggest there is a correlation between ROS levels returning to normal and stopping of escape behavior. But in all the previous data, the escape index % calculation necessitated removing individuals as soon as they started the escape behavior. If this argument is kept then there needs to be data describing variation in the timespan of escape behavior bouts.

6. The point about competition driving diversification is interesting, but there is no strong evidence for that trend in this paper. Lines 410-411 “competition statement” deserves testing (indeed, the mechanism described here may enable a formal test), but is not a strong/appropriate conclusion of this manuscript.

Minor comments:

1. Line 25: “the most fundamental and intriguing” appears to be hyperbole. There are many varieties of strong competition, including removal of hosts via death without superparasitism, and direct behavioral aggression/territoriality.

2. Line 44: “places”. Perhaps “induces” is a better fit

3. Line 68: “more likely”—do these papers really argue that symbionts are the more likely driver of manipulation, or that they’re simply another force to consider.

4. Line 95: “massive”—very frequent?

5. Line 114: “a wasp egg”—at least one wasp egg? Current wording is ambiguous and suggests a “single” egg.

6. Line 122: It looks like there was a very low level of escape behavior present in the males (Fig1d). If this is the case, then it was not “absent”.

7. The transition to focusing only on the venom RhoGAP proteins (line 151) is problematic. Later on the authors explain the connection with ROS and former studies linking ROS with escape behavior under anoxia. Briefly referencing the phenotypic connection at this earlier stage would be helpful.

8. Line 214: “triggering escape behavior”. The authors never perform experiments that only deliver EsGAP1, so they can’t say how it triggers the behavior. It would be more consistent to say that the locus had the strongest effect on reducing escape behavior under knockout conditions.

9. Line 336: “Repelling” = non ideal word choice. Passive avoidance via host manipulation = faster?

10. Line 340-42: Although I agree that superparasitism was likely a major selection force driving EsGAP gene evolved, this paper does not prove this without question and there are other possibilities. Is it possible that the ROS effect may have other benefits for the wasp, such as altering fat composition in the host (I believe the original anoxia behavior story had something to do with lipids in the fly neurons)? A macroevolutionary study comparing parasitoids with EsGAP genes, specifically looking at gene duplication levels and levels of ecological opportunity for superparasitism would be one way to test this explicitly.

11. Lines 391-393: It is not accurate to say that the generalist is “adapted” to a wide range of hosts, only that it can effectively manipulate their escape behavior.

12. Lines 391-403: The origins of the manipulation genes in the common ancestor are worth discussing, and I like the connections with papers on competition driving increased resource diversity. But the adaptive stories a bit speculative. There is no evidence whether the ancestral state was a generalist vs specialist, and so impossible to know whether there was an expansion of manipulation in Lh or a loss of abilities in Lb. And if the concurrent paper suggests a strong role of immunity, then that story needs to be elaborated on here. The current statement is vague.

I hope the authors find these recommendations useful, as it was a pleasure to read and review this work.

Respectfully,

Jesse N. Weber

Reviewer #3 (Remarks to the Author):

This manuscript reports on a very interesting study, consisting of a comprehensive set of experiments that show that, and how, parasitoids can manipulate the hosts’ behaviour in such a way that superparasitism is reduced. The strength of the paper is the level of detail at which the mechanisms are

elucidated, revealing a set of genes that is expressed almost exclusively in the venom gland; three of these genes possess RhoGAP domains, and silencing these genes eliminates the effects on host behaviour. The RhoGAP-containing proteins are detectable in the brain of the host within an hour of parasitization, and strongly elevate ROS-levels in the brain. This ROS elevation is subsequently associated with the hosts' behaviour, i.e. moving away from a patch after parasitism. Finally, this host behaviour manipulation can be induced in various host species, and a closely related parasitoid species possesses similar paralogous genes, that induce similar behaviours in its hosts.

This study is an exquisite example of manipulation of host behaviour by a parasitoid, and these studies are of high interest to a wide research community. One of the best researched examples of host behaviour manipulation is the Baculo-virus induced tree-top disease in caterpillars, which is associated with enhanced transmission ability of the virus. The current study is an intriguing example where the host behaviour manipulation by the parasitoid is associated with competition avoidance among conspecific parasitoids. The combined experiments are very convincing, including behavioural assays, functional characterization of the parasitoid genes and hosts responses, as well as an expansion of the study beyond the two-species interaction towards a phylogenetic analysis. I feel that these findings provide novel insights in host-parasitoid interactions, as well as the extent to which host behaviour manipulation can evolve as an adaptive strategy in parasites.

Although I do find the combined experiments highly convincing evidence for host behaviour manipulation by a parasitoid to avoid risks of superparasitism, I am much less convinced by the “eco-evolutionary context” in which the study is being placed. Both the introduction and the discussion contain a collection of rather broad statements that are often not fully accurate or clear. A fairly straightforward evolutionary scenario is conceivable, in which a parasitoid manipulates its hosts' behaviour to limit the risk of other parasitoids superparasitizing. However, the authors choose to link it rather speculatively to broad ecological concepts as community composition, host range expansion and speciation, which seem to go well beyond the collection of the experiments.

Firstly, the introduction starts with a paragraph on “how competition shapes communities”, and adaptations to new environments or conditions. In subsequent sections, superparasitism is also linked to host range expansion and speciation processes. Several of these statements are slightly off, in my opinion, or unclear. Moreover, superparasitism and the entire study focusses on intraspecific competition within a single parasitoid species. In this context, the ecological consequences are primarily at the level of population dynamics of these parasitoids (e.g., superparasitism does not affect host-parasitoid interactions as much as it affects conspecific parasitoid-parasitoid interactions), and competition with other parasitoid species is not addressed in this study. Therefore, the appropriate “wider context” is how manipulation of host behaviour can evolve as a strategy for parasites, and the potential consequences on the parasites' population dynamics. All the mentioned associations with community composition, host range expansion and adaptations to new environments are tenuous at best, and do not seem to be warranted by the focus of the manuscript.

Moreover, the theoretical presentation of superparasitism, superparasitism avoidance and host discrimination is somewhat convoluted and unclear. Firstly, there is a distinction between costs and benefits (of avoidance) for self-superparasitism and conspecific superparasitism, for both the first female that exploits a host and any subsequent females that may superparasitize or avoid this when they detect it is already parasitized (i.e., host discrimination). This is not being addressed clearly. Also, the authors focus on the benefits of superparasitism avoidance from the perspective of the superparasitizing females, while it seems evolutionary more plausible that this strategy would evolve when it has a benefit for the parasitoid that is manipulating the hosts' behaviour. When high costs or risks of superparasitism exist for the parasitoid that is the first to exploit a host (e.g., her chances of reproductive success are substantially diminished when the host is subsequently superparasitized by another female, either because the offspring of the second female takes over the host, or because the host has a higher likelihood of dying from the multiple infections), a strategy could evolve that manipulates the hosts' behaviour to minimize the risk of superparasitism. In contrast, when this strategy is being discussed as a means by which the "second" female avoids time wasting in host handling and host discrimination, it requires the assumption of a cooperative strategy between parasitoids, where the first females aids/saves the subsequent females from wasting time. This almost constitutes a group selection argument.

Another concern I have is that the level of detail in part of the methods is insufficient to enable others to repeat the experiments. For example, for host behaviour experiments, it is not clearly stated that multiple parasitoids (and how many) were searching on a patch simultaneously, and how the escape indices were calculated. Also, details on the collection of samples/tissues for the transcriptomics experiment on different life stages is not provided, nor on the different tissues for ROS expression profiling. Additionally, a few of the key findings are provided with qualitative descriptions, rather than rigorous statistical tests. For example, Lines 218 – 220 describe a crucial experiment to show the causality of the venom EsGAP protein in inducing ROS in the hosts' CNS, which then relates to the hosts' induced behaviour. However, it is only presented with a qualitative description and 1 'representative' image, and not with a statistical analysis. This part of the study requires more details and a substantive analysis of the patterns. Additionally, more details are needed for part of the methods (see below for specific suggestions).

Minor revisions:

L24 – 27: clarify that this is all about intraspecific competition to avoid confusion later on.

L40 – 47: consider strongly revising or omitting: (L43: not only when exposed to new environments; L44: the link between adaptations for competitive interactions and new conditions is far-fetched or not well explained; L46: there is substantial empirical research on intraspecific competition; L47: the link with speciation is far-fetched within the context of the present study).

L66 – 69: Unclear sentence. Here, the virus manipulates its hosts to enhance its own transmission. What is meant with “more likely’ (than what?) and what is meant by “remains adaptive” (for whom?).

L76: replace “bases” with “basis”?

L98: consider mentioning here already that the vast majority of larvae on the walls of the bottle turned out to be parasitized (fig1e), rather than unparasitized hosts that might simply be trying to escape parasitism. This is a stronger clue that this may be for “superparasitism avoidance” than the observation that female parasitoids prefer to search on the medium; the latter could simply indicate that larvae try to escape from being attacked by a parasitoid.

L 182 – 183: “most tissues”? It is unclear from the methods section which other tissues were dissected or examined.

L 218 – 220: this experiment is a very important part to show the causality of EsGAP in inducing ROS in the CNS, which then relates to the hosts’ induced behaviour. However, it is only presented with a qualitative description, and not with a statistical analysis. This part of the study requires more details and a substantive analysis of the patterns.

L 240 – 243: I believe CK means “not exposed to parasitoids”, so for each species, the numbers of eggs in the various species is not “in CK” but “for parasitoids with the control construct P_dsGFP”

L250: not clear how these results “strongly indicate” a similar role in other drosophilid parasitoids: they indicate that Lb can manipulate this behaviour in a range of host species.

L253: replace “relative” with “closely related”?

L 258: replace “modes” with “clades”?

L292: could this speculation on “LGT from protists” not be tested with a fairly simple phylogenetic analysis/comparison, by expanding the analysis for figure 5a with protists/prokaryotes?

L 313: should these results not be presented in the main text as part of figure 5? They are crucial to show that the host behaviour manipulation is also found in a related parasitoid species, and not an unique feature of L. boulardi.

L 321: I do not agree that it has been a “concern”: clear benefits and mechanisms have been proposed, such as the ability for host discrimination.

L 321 – 325: I suggest merging these two sentences on the ability of parasitoids for host discrimination into one

L 325: “infectious information” is not clear or accurate (the larvae are not infectious)

L 331 – 342: this entire section is confusing and partially incorrect. “Host discrimination” is not a real contrast to the “host behaviour manipulation” that is investigated in this study: host discrimination is a strategy that females may employ when they encounter already parasitized hosts (or unsuitable host species). The manipulation of host behaviour is a strategy that females may employ to reduce the risk of other females “finding” the host they have already exploited. The benefits of this strategy of host behaviour manipulation are not primarily for the female that may waste time or eggs by superparasitism, but for the female that has already made an actual investment in this host (deposited

an egg in it), and can so reduce the risk of other females exploiting this host as well (by superparasitization).

L 346: consider to present Figure 5c as a separate figure. It is not connected to figure 5a and 5b, and would benefit from a more extensive legend.

L361 - 364: I would consider it much more likely that these venom proteins are there to stop the host from differentiating lamellocytes (i.e., to try to sabotage the hosts' immune response), than to avoid damage to host neurons.

L391: what does "this strategy" refer to?

L 396 – 398: Too broad statement. Competition can also lead to strong specialization or a reduced realized niche (compared to the fundamental niche). Its relation with host range expansion has no good foundation.

L 402: "the recent study" is a bit confusing. It is not the current study, but a related study? I also do not fully understand this statement.

L 406: remove the part "that can be recognized by the wasp". The most parsimonious explanation is that it "induces behaviour in the host to limit [the risk of] superparasitism".

L 410 – 411: I fail to see how this study "provided key empirical evidence for how [...] it structures ecological communities at long time scales". Suggest removal of the latter part of the sentence, or to provide clarification.

L435: please, rephrase and explain "Genes are displayed based on the TPM of venom".

L 448 & L 453: "The dashed line" instead of "Dashed lines"?

Figure 4a: this figure suggests that the number of hosts that were "without eggs" was also higher in PdsEsGAP1? Did you test for this?

Legend Figure 5a: please explain both the circle and asterisk in the legend of the figure

L 497- 500 / Legend Figure 5c: I believe the proposed model is that this is a strategy, not to combat their host, but to "avoid superparasitism (or competition) by conspecifics. Replace "leading to" by "inducing", and I suggest removing "and subsequent superparasitism avoidance". The sentence on long-term effects, host shift or expansion or speciation seem not warranted or directly related to the findings.

L 506: too little information to repeat the rearing conditions (e.g. quantities and % of apple juice agar)

L 512: How many parasitoid females were simultaneously released on a host patch?

L 512 L518: Replace "Well-mated" with "Mated": females are either mated or not

L 519: what is the "immediate repellency behaviour" and is this similar to the observed escape behaviour after parasitism?

L 524: how were the escape indices calculated?

L 531 – 547: consider moving this section to later, to maintain the same order as in the results.

L540: Replace "Figure 2b and 2d" with "Figure 3"

L572: provide details on the collection of the different life stages and venom for transcriptome analysis

L 587 & L589: write out the abbreviations KO and TMP

L 601: I am no expert on protein analysis, but why was only the supernatant analyzed and not the pellet?

L 634: Rephrase to something like "Genes in the venom with an expression of 7.3 transcripts per million (TPM) were"

L 635 - 637: Rephrase and clarify

L 686: Clarify "Reactions". Is this including the cDNA synthesis? If not, please provide these details.

Reviewer #1 (Remarks to the Author):

The Authors present a long read assembly of the Hymenopteran parasitoid of *Drosophila*, *Leptopilina boulardi*. The assembly itself is a great improvement from a currently available one on NCBI: scaffold n50 of 458kb vs the authors new assembly with a contig n50 of 3.6 Mb. Furthermore, that assembly appears to be very complete by both CEGMA and BUSCO. Unfortunately the authors do not provide (that I could find) which taxonomic level the analyses used (insecta or hymenoptera?). This can have a dramatic effect on assembly “completeness”, especially in *Leptopilina*. This is a minor concern, and does not detract from the paper’s conclusions or impact.

The original BUSCO assessment was performed using Insecta (odb9). We have also performed another BUSCO assessment using the hymenoptera set (odb9) as suggested, resulting in 95.4% and 91.8% as partial and complete, respectively. Related information has been added in the updated Supplementary Table 1.

In parasitization assays, the authors observed host larvae leaving food and found that the majority of larvae were parasitized. Furthermore the fly larvae that didn’t leave the food suffered high mortality, presumably due to superparasitization. They hypothesize that the host leaving food could be superparasitization avoidance. Injected wasp venom elicit the same response, while mock injections do not. This indicates that the wasps are behaviorally manipulating their hosts. From the wasp assembly and venom annotation, there were three highly expressed and venom specific gene products containing RhoGap domains, thus the authors hypothesized these venoms could be responsible for the host manipulation. RNAi knockdown of the RhoGap containing genes confirms this hypothesis. It is surprising that all three knockdowns have a greater than 50% effect on their own. This could be due to threshold effects or perhaps non-specific knockdown of the other paralogues, which is not tested in the diagnostic RT-PCRs (Supp. Fig 2a and 6a). If the alternative knockdown controls were not done, this should be mentioned.

This is an important point which was also raised by another reviewer. All the three genes are of short gene length and mainly occupied by the RhoGAP domain, leaving no space for us to design specific targets for the RNAi experiment. As per your suggestion, we have additionally tested the expression levels of the other two genes upon the injection of one of the three genes and indeed detected reduced expression in non-target genes. We have mentioned these results in both main text (lines 170-171, 330-331) and new Supplementary Figs. 3b and 9. We note that these spillover effects do not change our main conclusion regarding the link between these newly evolved RhoGAP genes and the escaping behavior, and that fine differentiation of functions among these genes is not the purpose of this study.

The authors show that parasitization, and venom specifically, induces ROS in the larvae CNS. ROS is directly involved in escape behavior as shown by flies genetically manipulated to suppress ROS also suppress escape behavior in the presence of wasps. RhoGap is directly implicated in CNS ROS accumulation by immunohistochemistry showing CNS localization and RhoGap knockdown in wasps preventing ROS accumulation. The authors attempt to address the ecological implications of wasp induced host “escape” behavior by assessing the number of superparasitized fly larvae

when wasps have RhoGap knocked down. Reduction in RhoGap leads to reduced “escape” and increased number of eggs per larvae. They go further to characterize this effect in other *Drosophila* species, showing that the wasps are manipulating a conserved mechanism of fly behavior (possible hypoxia avoidance). The phylogenetic analysis and gene family reconstruction of RhoGap venoms is perhaps the weakest and or least clear section.

We appreciate your overall positive comments on our study and agree that the previously evolutionary analyses are not ideal. This section has been improved based on your suggestions. Please see detailed responses as below.

I have several concerns: (1)The blastP search strategy for RhoGap homologues resulted in very hymenoptera/hexopoda weighted sampling. This most likely leads to problems with long branch attraction, inflating the evidence of lateral gene transfer, especially for fast evolving genes like the venoms are proposed to be. If the tree topology is to be believed, the Leptopilina RhoGaps originated from 6 independent LGTs. Likewise, there are several other long branched hymenopteran and hexapod RhoGaps placed with the non-metazoans. It is much more likely that these long branched genes arose from duplications within hexapod or hymenopteran evolution, evolved rapidly, and are erroneously placed deep in the tree because of it. A search strategy including the conserved Leptopilina RhoGap genes would fill out the tree more evenly because it wouldn't rely on identity to a fast evolving gene. Perhaps an even better approach would be to pull available RhoGap sequences from the InterProScan database, subsample and attempt to align them with the Leptopilina RhoGap sequences.

We did recognize that LGT is not the only possible origin of these RhoGap genes and indeed claimed the possibility of a duplication origin of these genes in the original manuscript (please see Fig. 5c and lines 292-294, 295, and 367 of our original manuscript). We never claim the LGT scenario alone. To this respect, we stand in the same front.

We agree with your concern that the current sampling strategy is not ideal, but we think this is still the most manageable approach we can find. We stress that RhoGap is an extremely large gene families with ultra-widespread distribution across Eukaryota. As we explained in the original manuscript, the BLASTP search reported more than 10k hits across all eukaryotic hits. Actually, the BLASTP search did not recover too weighted sampling to Leptopilina, e.g., it returned more hits from Diptera (781) than those from Hymenoptera (520). However, the distribution of sampling was indeed uneven, which was mainly caused by the difference in artificial interests across different scientific communities. Given that hits were not evenly distributed and that it is unpractical to perform a phylogenetic analysis using all sequences, we independently downsampled based on BLAST e-value for each subgroup of Eukaryotic species, i.e., different subgroups with increased evolutionary relationship to Leptopilina (the organisms of Parasitoida, Hymenoptera excluding Parasitoida, Hexapoda excluding Hymenoptera, Deuterostomia, and Metazoa excluding Deuterostomia or Hexapoda). We meant to use this strategy to balance that the analysis being doable and that the sampling being as representatively and evenly as possible.

We have also followed your another suggestion that using all sequences available in InterPro. We were able to retrieve 32,928 sequences with the domain (IPR000198). Although we have made an attempt to include all sequences in the phylogenetic analysis, the resulted tree is a mess as expected. In the tree, most topologies are of poor bootstrap supports and our focused EsGAP genes are accompanied with longer branches and clustered with other long branches. That is, the issue of long branch attraction (LBA) was further magnified, possibly due to the great divergence between these novel RhoGAP genes and those typical ones. We further used CD-HIT to cluster sequences sharing >95% identity, which resulted in 9917 relatively unique sequences. However, the phylogenetic analysis using this downsampled set was still problematic with issues of LBA and poor bootstraps. Overall, such analyses with so many divergent sequences cannot result in reliable, informative topologies.

Anyway, we have removed the tree across all animal species from the main slot, added a tree focusing on Hymenoptera species as another supplementary figure, and additionally mention the LBA effect in main text. We hope these revisions might make this section less confused.

(2) In Figure 5a, if the Lh RhoGap sequences were also included (in a more evenly sampled tree), it would help to identify when the LGT or duplication happened. Leaving these out doesn't make sense to me and leads to unnecessary reader questions.

We apologize for the unclear figure legend and inconsistent labels within the same figure that might cause your confusion. Actually, Lh sequences had been included in original Fig. 5a. Note that those red branches indicate Leptopilina species, including both Lb and Lh. Here, the color usage is different with that in original Fig. 5b, in which red and blue branches indicate Lh and Lb, respectively.

Anyway, a newly designed Fig. 5 has been presented. The broader trees have been moved to supplemental, avoiding the issue of inconsistent color usage.

(3) Using *Microplitis demolitor* (misspelled on line 495) as the outgroup explicitly makes Fig 5b contradict the placement of the RhoGap gene family expansions in 5a. 5a shows them outside of hymenoptera (*) while 5b shows them within hymenopteran RhoGaps. If the authors wish to maintain that the venom RhoGaps could have come from a LGT, this tree should be unrooted so that it doesn't start with the assumption that all Leptopilina RhoGaps could coalesce as expected by the species tree (no LGT required). Without branch lengths, node support, and more outgroup sequences, Figure 5b supports duplication leading to the RhoGap venoms without having to hypothesize the less likely and poorly supported LGT from a distantly related eukaryote. I recommend removing figure 5a because it poorly supports LGT due to methodological issues. I also recommend that 5b should include several other hymenoptera RhoGaps (evenly sampled) and use a well annotated insect (like fly) as an outgroup. This new tree with node support would be far superior to 5a. If the Lb and Lh species specific expansions do not cluster with the rest of the Leptopilina RhoGaps (or hymenoptera), this would be much better evidence of LGT, and adding more diverse sequences would be warranted. Without these changes, proposing LGT as a possible origin of the venoms is completely unnecessary, when duplication and

neofunctionalization are a likely and supported hypothesis.

Fig. 5b was not meant to discuss LGT or other phylogenetic origins. Instead, this Leptopilina tree was only used to present Leptopilina RhoGAP genes with a biological meaningful order, i.e. distinguishing these lineage-specific genes from those orthologous ones. The point of this panel is the agreement between the phylogenetic divergence and the divergent pattern of expression.

Regarding your request, we have newly added a phylogenetic tree outside Leptopilina, but focusing on Hymenoptera. The newly added tree (Supplementary Fig. 8) is presented in both forms of cladogram (with bootstrap) and phylogram (with branch length). This tree presents clear evidence of host escape-related genes forming a distant, monophyletic sublineage. Also, this tree is not integrated with the Leptopilina-only tree, which we have other purposes as explained above.

Finally, the authors mention that the species specific RhoGap domains are accompanied by several other domains, but the conserved RhoGaps are not. The identity of the extra domains for Lb and Lh would be interesting to mention in this paper. Since Lb and Lh RhoGap venoms come from a single Leptopilina ancestor and seem to induce ROS and escape behavior, do they have the same domain architecture? Do all of the venom RhoGaps have the same extra domain? Have some of the duplicated venom genes gained different domains?

Actually, the pattern we reported is that the lineage-specific RhoGap genes lack of additional domains, while the conserved ones are accompanied with other domains (see Fig. 5b and lines 263-267 and 275-280 of our original manuscript as well as Fig. 5a and lines 280-284 and 290-296 of our revised manuscript). Please note that the bars of original Fig. 5b indicate the ratios of the domain length to the entire length of gene models, i.e. the longer bar, the shorter gene length (without additional domains).

We recognize that the original presentation might cause similar confusion to other audience, we therefore have updated this figure by directly displaying the inside domains and the actual gene length (new Fig. 5a).

I believe the work presented here is given sufficient detail to be reproduced. As far as I can tell the statistical tests are appropriate. Fig 3b and 3d could use a mixed model to gain more power, however the t-test is conservative, and the point is made. There are no stats for the time series larvae escape figures (1d and 4c). These figures are convincing by themselves, but stats would be nice.

All of these statistic issues have been appropriately fixed. We are happy to report that they are still significant based on the updated statistical analyses (see new Fig. 3b and 3d, along with text in lines 493-494, 499; newly added Supplementary Tables 4, 5 for detailed significance values of each time point in Fig. 1d and 4c, along with text in lines 471-472; lines 537-539, 814-818).

In general, the authors work presented here is novel and of great benefit to the research community. They tell a full genotype to phenotype story WITH evidence of implications on host/parasite selective forces, which is sorely lacking in the field. I believe this paper is very strong and recommend publication if either (A) the LGT hypothesis and Figure 5a are deemphasized/removed or (B) a more robust analysis is added which supports LGT.

We appreciate your high evaluation on our study again. As explained above, the origin of RhoGAP genes is fairly discussed and the original Fig. 5a has been deemphasized by moving to supplemental and adding with supports from an independent analysis including all Hymenoptera sequences.

Reviewer #2 (Remarks to the Author):

Summary

This paper elegantly addresses the ecological, genetic, and physiological basis of an extended phenotype; namely, it shows that evolutionarily derived venom proteins in a parasitoid wasp induce a behavioral shift in its host--*Drosophila* larvae. This shift is facilitated in part through venom-related induction of ROS in the host nervous system. The ecological function of the manipulation is that it lowers levels of superparasitism under lab conditions, likely leading to decreased competition for developing wasp larvae. The compelling mechanistic story explaining manipulation is supported by a diverse set of analyses, including broad infection and behavioral time courses, multiple "omics" approaches, genetic manipulation, histology, and phylogenetics. As it stands, most of the results in this paper are compelling and original, and all the statistical, phylogenetic, and molecular methods are appropriate. I commend the authors for offering such a well written and thorough manuscript. I believe the results will prove interesting to a wide readership. However, there are a few areas where additional description of results/methods would be helpful. I also suggest places where it seems prudent to adopt more cautious/nuanced interpretations of the ecological and fitness results. Finally, note a few irregularities/omissions in data that need to be explained.

We appreciate your high evaluation and kindly suggestions on our study. We understand your concerns with partial interpretations of the original manuscript. Please see detailed responses as below.

Main comments:

1. -A main area of concern is that the authors provide little quantification of the negative fitness effects of superparasitism for the wasp. The issue of wasp fitness only seems to be addressed when considering those larvae that did not exhibit any escape behavior through the full course of each experiment. The authors speculate that the 74% mortality column in Fig 1f results from super parasitized hosts. The actual distribution of eggs per host per behavioral category (escape vs not escape) is not reported. Moreover, the number of wasps per host is not described for those wasps that do hatch (Fig 1e,f). If multiple wasps come out of larvae that are expressing the escape behavior, then it is only removing the selection from "excessive" superparasitism. If previous studies more fully quantified the fitness impact of competition via superparasitism in this system, then it needs to be included in the discussion.

We apologize that the incomplete explanation of solitary parasitoids might cause the confusion. Superparasitism generally causes premature death of parasitized hosts and consequently stops the development of laid parasitoids (e.g., see the newly added Supplementary Fig. 1). In addition, the study system in this study (*Leptopilina*) is a typical solitary parasitoid. Solitary parasitoids usually lay a single egg inside the host per each oviposition. Even multiple times of oviposition occur, only one of the laid offsprings can successfully develop into the adult and emerge from a single host (if still survived). Thus, the main negative fitness effect of superparasitism for solitary parasitoids is the absolute lethality of other offsprings within the same host. This strictly exclusive pattern makes this system an ideal model to address

superparasitism, because behavioural qualification of “superparasitism” is definite here: the appearance of more than one egg in a single host is simply defined as superparasitism, without the further issue of “excessive” superparasitism. To avoid causing confusion for audiences, we have clarified the background of solitary parasitoids at the beginning of results (see lines 80-85 of our revised manuscript).

In addition, we agree with your point that we should compare and present the laid eggs between escaped and non-escaped hosts to demonstrate the association between the high mortality and superparasitism. These results, along with the premature death images of superparasitized hosts, have been presented in newly added Supplementary Figs. 1 & 2 and appropriately mentioned in main text (see lines 117-118 and 120-123 of our revised manuscript).

There is also a discordance between Fig1a and 4a. The original result shows a small but non-zero level of superparasitism at 60min, but this appears to vanish when the authors begin contrasting host behavior when exposed to control vs EsGAP1-knockdown wasps. It also seems that the interpretation of superparasitism levels is somewhat contingent on the exposure time to parasitoids. For example, in Fig1a almost all of the hosts experiencing superparasitism at 4hr. Perhaps the disconnect with fig 4a is the ratio of flies to wasps? The methods suggest that 50 dsRNA-injected wasps were used for experiments, but the ratio of hosts-wasps is not explained.

We apologize for the incomplete information in the original manuscript. Method information of behavior assays have been extended in the revised manuscript (see lines 591-596).

Regarding your concern with the difference in superparasitism ratios at 60 min between wildtype (Fig. 1a) and control (dsGFP) (Fig. 4a) parasitoids, both behavior assays were both performed under the same ratio of hosts to wasps (1:10), i.e. the discordance was not caused by the different assay condition. Instead, we think the most possible explanation is the different generations of Lb being used in the two assays, respectively. It is not uncommon to see slightly behavioral changes, particularly some occasional behaviors, between different generations of insects. We note that the wildtype parasitoids maintained a near-zero level of superparasitism ratio until 45 min and got a slightly increased level at 60 min (see Fig. 1a), i.e. the 60 min after exposure to parasitoids is a possible boundary point when a very few superparasitism cases begin to occur. Unlike the dynamic statistics along different time points in the assay of Fig. 1a, behavioral assay of Fig. 4a was performed at a single time point (60 min). Thus, the slight behavioral difference around the “boundary” time between different generations of wildtype/control parasitoids might cause the discordance as you pointed out. In addition, the total observed size is different between the assays of Figs. 1a (n=285) and 4a (n=187), which might also cause slight difference in background-level statistics. Since both superparasitism ratios of Figs. 1a and 4a are significantly and obviously lower than those of other groups, the slight difference between each other has very little effect on our main conclusions regarding the increased superparasitism ratios along the time after exposure (Fig. 1a) and the RNAi effect of dsEsGAP1 (Fig. 4a).

2. Another point that could benefit from added ecological context concerns the origins of the long-term lab origins/propagation of the wasp strains. The *L. boulardi* (Lb) wasp strain appears to have been isolated in 1978, and the strain of the other species has been maintained in the lab since 2002. If the genome sequences came from this lab strain, was any attempt made to confirm that the EsGAP paralogs exist in wild caught wasps? And if so, are there same number of copies between lab and wild? Some of the earliest studies of large effect behavioral adaptation genes were found to be either only observed in the lab (e.g., rover/sitter polymorphism in *Drosophila*), or were actually driven by lab-induced selection (e.g., *npr1* induced clumping during foraging in *C. elegans*). While the presence of venom-specific EsGAP genes in both the generalist and specialist parasitoids suggest their common ancestor must have had at least one copy, it isn't hard to envision how superparasitism selection could be much stronger under lab conditions than in the wild, and that some of the molecular evolution reported here may be lab derived. This is particularly important given that the authors repeatedly suggest the mechanisms revealed in this study are relevant to naturally occurring superparasitism.

This is an important point. We agree that we cannot rule out the impact of lab-derived selection on the molecular evolution of this gene family. Correspondingly, we found tandem copies of EsGAP genes sharing near-identical sequences in both Lh and Lb genomes, indicative of very recent duplication events. We appreciate your idea that making comparison of EsGAP paralogs between wild-caught and lab maintained wasps. However, the wild samples are currently unavailable. Lb strain was initially isolated in Brazzaville, Congo and Lh strain was isolated in California, USA. We have not sampled these two species in local field during the past five years, although we were able to collect other sister species of *Leptopilina*. We think the geographic isolation might lead to speciation of *Leptopilina*, and that Lh and Lb do not distribute in China. We also note that these two lab strains are commonly used by most laboratories engaging in *Leptopilina*. We have contacted with some international collaborators and got negative feedbacks regarding the availability of wild-caught samples. Upon the relief of Covid-19 epidemic around the world, we hope we have the opportunity to catch wild wasps in the near future.

Anyway, we have added text to mention the potential impacts from lab condition to clarify this issue (see lines 426-432).

3. There is a worrisome irregularity about the number of RhoGAP domain transcripts/proteins that are highly expressed in Lb venom. Only 3 loci are dissected in the knockdown experiments and reported as being highly expressed. But Fig5b show 4 highly expressed paralogs, and SupTable 3 shows a 4th gene (LB2_216_152) with the domain and the same high length ratio. It seems unlikely that this 4th locus would have been excluded from preliminary mechanistic experiments, but if so why? Even if preliminary results no effect of LB2_216_152 on ROS or behavior, those data should still be reported as they speak to the varying effects of that protein domain.

We apologize for the confusion. The four highly expressed paralogs showed in original Fig. 5b are LB2_151_037 (*EsGAP2*), LB2_330_135 (*EsGAP3*), LB2_255_017 (*EsGAP1*) and another

tandem copy of *LB2_255_017 (EsGAP1)* sharing the identical sequence. To avoid confusion, we have renamed the fourth gene as *EsGAP1'*. Thus, we cannot perform specific experiments respectively targeting on *EsGAP1* and *EsGAP1'*. The expression level of “the fourth gene” that you mentioned (*LB2_216_152*) is pretty low and lacks of proteome evidence. This gene was therefore excluded from the mechanistic experiments. Related information had been mentioned in the footnote of original Supplementary Table 3. To avoid causing further confusion to other audience, we have mentioned *EsGAP1'* in the legend of the main figure too (see lines 547-549).

4. Lines 161-171. The RNAi knockdown data, including the qRT-PCR and behavioral measurements, are interesting and integral to the paper. However, there is a worry that the dsRNA oligos for each locus might not have had a specific effect. The authors only report data for individual dsRNA-EsGap pairs. It is important to know whether the extra reduction in escape behavior from knockdown of *LB2_255_017* is due to the phenotypic effect of only that locus, or whether the *LB2_255_017*-dsRNA has a spillover effect that also reduces expression of the other two EsGap genes. The specificity of RNAi knockdowns varies greatly among species, and spillovers have been observed even with substantial sequence differences between loci. This speaks to the genetic architecture of the induced host phenotype (i.e., additive vs redundancy vs big+small effect loci).

This is point was also raised by another reviewer. All the three genes are of short gene length and mainly occupied by the RhoGAP domain, leaving no space for us to design specific targets for knockdown. We have additionally tested the expression levels of the other two genes upon the injection of one of the three genes and indeed detected reduced expression in non-target genes. The spillover effect has been mentioned in both main text (see lines 170-171 and 331) and new Supplementary Figs. 3b and 9. We note that these spillover effects do not change our main conclusion regarding the link between these newly evolved RhoGAP genes and the escaping behavior, and that fine differentiation of functions among these genes is not the purpose of this study.

5. Lines 186-188: the authors suggest there is a correlation between ROS levels returning to normal and stopping of escape behavior. But in all the previous data, the escape index % calculation necessitated removing individuals as soon as they started the escape behavior. If this argument is kept then there needs to be data describing variation in the timespan of escape behavior bouts.

We have added dynamic data across different time points to new Supplementary Fig. 2b and mentioned these results in main text (lines 198-199 of our revised manuscript).

6. The point about competition driving diversification is interesting, but there is no strong evidence for that trend in this paper. Lines 410-411 “competition statement” deserves testing (indeed, the mechanism described here may enable a formal test), but is not a strong/appropriate conclusion of this manuscript.

The point that competition drives diversification was proposed in previous studies. Indeed, this study does not present direct evidence to link the competition with speciation. Our original thoughts are to establish a possible and reasonable linkage between the intraspecific competition behaviors (superparasitism avoidance) in both species and the ecological contexts of Lb and Lh (one specialist and one generalist). Anyway, we have modified this part by toning down related points and removing strong statements (see lines 435-450).

Minor comments:

1. Line 25: “the most fundamental and intriguing” appears to be hyperbole. There are many varieties of strong competition, including removal of hosts via death without superparasitism, and direct behavioral aggression/territoriality.

We have modified this as “one of the most fundamental and intriguing intraspecific competition behaviors”.

2. Line 44: “places”. Perhaps “induces” is a better fit

It has been changed as suggested.

3. Line 68: “more likely”—do these papers really argue that symbionts are the more likely driver of manipulation, or that they’re simply another force to consider.

These references raised an alternative force to consider. To avoid confusion, we have modified this sentence as “suggesting that superparasitism is likely to be manipulated by symbionts as an adaptive strategy for virus transmission”.

4. Line 95: “massive”—very frequent?

It has been modified as suggested.

5. Line 114: “a wasp egg”—at least one wasp egg? Current wording is ambiguous and suggests a “single” egg.

We apologize for the confusion and have modified it as suggested.

6. Line 122: It looks like there was a very low level of escape behavior present in the males (Fig1d). If this is the case, then it was not “absent”.

We have modified the original sentence as “we did not find frequent host escape” to avoid confusion.

7. The transition to focusing only on the venom RhoGAP proteins (line 151) is problematic. Later on the authors explain the connection with ROS and former studies linking ROS with escape behavior under anoxia. Briefly referencing the phenotypic connection at this earlier stage would

be helpful.

After discussion with our colleagues, we are still satisfying with the current context. Our focuses were indeed attracted by the overrepresentation of RhoGAP proteins in the significantly highly-expressed list. Of course, we should limit their roles in venom but extending to the host escape behavior. We have accordingly modified the sentence by proposing the role of RhoGAP proteins in the venom instead of directly linking with host escape (see lines 159-160). However, previous functional information related to RhoGAP is not enough to lead us to relate it with escape behavior; the relatedness is marginal. Our subsequent RNAi experiments convince us the role of these RhoGAP genes in inducing host escape. Moreover, these experiments covered all significantly highly expressed genes, rather than testing the three RhoGAP genes only. That is, the incline to RhoGAP at this time point did not bias our further surveys toward them.

8. Line 214: “triggering escape behavior”. The authors never perform experiments that only deliver EsGAP1, so they can’t say how it triggers the behavior. It would be more consistent to say that the locus had the strongest effect on reducing escape behavior under knockout conditions.

We have changed this sentence as suggested.

9. Line 336: “Repelling” = non ideal word choice. Passive avoidance via host manipulation = faster?

We have changed this sentence as “Rejecting parasitized hosts via manipulating them to escape ...”.

10. Line 340-42: Although I agree that superparasitism was likely a major selection force driving EsGAP gene evolved, this paper does not prove this this without question and there are other possibilities. Is it possible that the ROS effect may have other benefits for the wasp, such as altering fat composition in the host (I believe the original anoxia behavior story had something to do with lipids in the fly neurons)? A macroevolutionary study comparing parasitoids with EsGAP genes, specifically looking at gene duplication levels and levels of ecological opportunity for superparasitism would be one way to test this explicitly.

We agree with your point that other effects of ROS might place selection on EsGAP genes too. We have mentioned this possibility in main text (see lines 421-427). Regarding your further suggestion, it is currently impossible to perform a macroevolutionary survey in the near future, because we do not know the superparasitism-related background of the parasitoids whose genomes are unavailable. We do not have these species in hand either. This might be an interesting, long-term project by whole-genome sequencing and “phenotyping” a predesigned set of representative parasitoid species.

11. Lines 391-393: It is not accurate to say that the generalist is “adapted” to a wide range of hosts, only that it can effectively manipulate their escape behavior.

Here, “adapted” means that the generalist species, Lh, has been able to parasitize additional species than *D. melanogaster*. To avoid confusion, we have changed “adapted” to our own meanings (see lines 439).

12. Lines 391-403: The origins of the manipulation genes in the common ancestor are worth discussing, and I like the connections with papers on competition driving increased resource diversity. But the adaptive stories a bit speculative. There is no evidence whether the ancestral state was a generalist vs specialist, and so impossible to know whether there was an expansion of manipulation in Lh or a loss of abilities in Lb. And if the concurrent paper suggests a strong role of immunity, then that story needs to be elaborated on here. The current statement is vague.

We understand your concerns. As explained above, our original statement was placed as one of the most likely scenarios that might link the observed superparasitism avoidance traits with the ecological contexts of these two species. We have reorganized this paragraph (see lines 435-450) by removing those strong statements relying on ancestral status before speciation.

I hope the authors find these recommendations useful, as it was a pleasure to read and review this work.

Thank you. Your comments and kindly suggestions help improve the manuscript greatly.

Respectfully,
Jesse N. Weber

Reviewer #3 (Remarks to the Author):

This manuscript reports on a very interesting study, consisting of a comprehensive set of experiments that show that, and how, parasitoids can manipulate the hosts' behaviour in such a way that superparasitism is reduced. The strength of the paper is the level of detail at which the mechanisms are elucidated, revealing a set of genes that is expressed almost exclusively in the venom gland; three of these genes possess RhoGAP domains, and silencing these genes eliminates the effects on host behaviour. The RhoGAP-containing proteins are detectable in the brain of the host within an hour of parasitization, and strongly elevate ROS-levels in the brain. This ROS elevation is subsequently associated with the hosts' behaviour, i.e. moving away from a patch after parasitism. Finally, this host behaviour manipulation can be induced in various host species, and a closely related parasitoid species possesses similar paralogous genes, that induce similar behaviours in its hosts.

This study is an exquisite example of manipulation of host behaviour by a parasitoid, and these studies are of high interest to a wide research community. One of the best researched examples of host behaviour manipulation is the Baculo-virus induced tree-top disease in caterpillars, which is associated with enhanced transmission ability of the virus. The current study is an intriguing example where the host behaviour manipulation by the parasitoid is associated with competition avoidance among conspecific parasitoids. The combined experiments are very convincing, including behavioural assays, functional characterization of the parasitoid genes and hosts responses, as well as an expansion of the study beyond the two-species interaction towards a phylogenetic analysis. I feel that these findings provide novel insights in host-parasitoid interactions, as well as the extent to which host behaviour manipulation can evolve as an adaptive strategy in parasites.

We appreciate your high rates on our study.

Although I do find the combined experiments highly convincing evidence for host behaviour manipulation by a parasitoid to avoid risks of superparasitism, I am much less convinced by the "eco-evolutionary context" in which the study is being placed. Both the introduction and the discussion contain a collection of rather broad statements that are often not fully accurate or clear. A fairly straightforward evolutionary scenario is conceivable, in which a parasitoid manipulates its hosts' behaviour to limit the risk of other parasitoids superparasitizing. However, the authors choose to link it rather speculatively to broad ecological concepts as community composition, host range expansion and speciation, which seem to go well beyond the collection of the experiments.

We agree with your thoughtful criticisms and have carefully modified the manuscript in places. In the revised manuscript, many unsupported statements have been either toned down or removed. Overall, we only retain the ecological context of INTRASPECIFIC competition in Introduction. Content regarding speciation has been removed from the manuscript, while that regarding host range expansion has been limited to Discussion with appropriate contexts. Please see our specific responses as below.

Firstly, the introduction starts with a paragraph on “how competition shapes communities”, and adaptations to new environments or conditions. In subsequent sections, superparasitism is also linked to host range expansion and speciation processes. Several of these statements are slightly off, in my opinion, or unclear. Moreover, superparasitism and the entire study focusses on intraspecific competition within a single parasitoid species. In this context, the ecological consequences are primarily at the level of population dynamics of these parasitoids (e.g., superparasitism does not affect host-parasitoid interactions as much as it affects conspecific parasitoid-parasitoid interactions), and competition with other parasitoid species is not addressed in this study. Therefore, the appropriate “wider context” is how manipulation of host behaviour can evolve as a strategy for parasites, and the potential consequences on the parasites’ population dynamics. All the mentioned associations with community composition, host range expansion and adaptations to new environments are tenuous at best, and do not seem to be warranted by the focus of the manuscript.

In the revised manuscript, we have reorganized the paragraph to start with a context of how parasites manipulate the host behavior for their own benefit, and weakened the associations with community composition, host range expansion and adaptations to new environments in INTRODUCTION (see our newly organized Introduction). We totally agree to limit the ecological consequence of our findings to “intraspecific competition” but a whole community. Other statements regarding speciation have also been removed from the revised manuscript. We only retain, but substantially tone down, the extended discussion regarding the potential long-term effect of host range expansion. Indeed, previously studies have suggested the role of intraspecific competition in driving diversification evolution, which matches the ecological contexts of Lb and Lh as we newly discussed in main text (see lines 435-450).

Moreover, the theoretical presentation of superparasitism, superparasitism avoidance and host discrimination is somewhat convoluted and unclear. Firstly, there is a distinction between costs and benefits (of avoidance) for self-superparasitism and conspecific superparasitism, for both the first female that exploits a host and any subsequent females that may superparasitize or avoid this when they detect it is already parasitized (i.e., host discrimination). This is not being addressed clearly. Also, the authors focus on the benefits of superparasitism avoidance from the perspective of the superparasitizing females, while it seems evolutionary more plausible that this strategy would evolve when it has a benefit for the parasitoid that is manipulating the hosts’ behaviour. When high costs or risks of superparasitism exist for the parasitoid that is the first to exploit a host (e.g., her chances of reproductive success are substantially diminished when the host is subsequently superparasitized by another female, either because the offspring of the second female takes over the host, or because the host has a higher likelihood of dying from the multiple infections), a strategy could evolve that manipulates the hosts’ behaviour to minimize the risk of superparasitism. In contrast, when this strategy is being discussed as a means by which the “second” female avoids time wasting in host handling and host discrimination, it requires the assumption of a cooperative strategy between parasitoids, where the first females aids/saves the subsequent females from wasting time. This almost constitutes a group selection argument.

This is an important point. We are grateful for your thoughtful criticisms that help clarify the

ecological consequence of superparasitism avoidance. We have added a new paragraph to address the relationships among superparasitism (involving both self-superparasitism and conspecific superparasitism), superparasitism avoidance, and host discrimination (see lines 353-366).

Another concern I have is that the level of detail in part of the methods is insufficient to enable others to repeat the experiments. For example, for host behaviour experiments, it is not clearly stated that multiple parasitoids (and how many) were searching on a patch simultaneously, and how the escape indices were calculated. Also, details on the collection of samples/tissues for the transcriptomics experiment on different life stages is not provided, nor on the different tissues for ROS expression profiling. Additionally, a few of the key findings are provided with qualitative descriptions, rather than rigorous statistical tests. For example, Lines 218 – 220 describe a crucial experiment to show the causality of the venom EsGAP protein in inducing ROS in the hosts' CNS, which then relates to the hosts' induced behaviour. However, it is only presented with a qualitative description and 1 'representative' image, and not with a statistical analysis. This part of the study requires more details and a substantive analysis of the patterns. Additionally, more details are needed for part of the methods (see below for specific suggestions).

We apologize for the incomplete information of some methods. In the original manuscript, we mainly introduced the ratio of hosts to wasps for behavioral assays. The exact number of hosts in each test has been added in each corresponding figure legend. We have also extended the Method section with details to address your other concerns (see lines 568-576 for insects rearing conditions, lines 593-594 and 601-602 for the number of wasps and hosts in behavioral assays, lines 656-667 for the sample information of transcriptome, and lines 779-781 for the tissues to test ROS). As per your suggestion, we have also performed the quantitative analysis with statistics for the experiment of EsGAP protein inducing ROS (see lines 228-234 and new Fig. 3j).

Minor revisions:

L24 – 27: clarify that this is all about intraspecific competition to avoid confusion later on.

We have specifically limited all competition-related content as intraspecific competition.

L40 – 47: consider strongly revising or omitting: (L43: not only when exposed to new environments; L44: the link between adaptations for competitive interactions and new conditions is far-fetched or not well explained; L46: there is substantial empirical research on intraspecific competition; L47: the link with speciation is far-fetched within the context of the present study).

This paragraph has been substantially revised and integrated with the following paragraph (see lines 41-53).

L66 – 69: Unclear sentence. Here, the virus manipulates its hosts to enhance its own transmission. What is meant with "more likely' (than what?) and what is meant by "remains adaptive" (for whom?).

This sentence has been reorganized as “..., suggesting that superparasitism is likely to be manipulated by symbionts as an adaptive strategy for virus transmission”.

L76: replace “bases” with “basis”?

It has been modified as suggested.

L98: consider mentioning here already that the vast majority of larvae on the walls of the bottle turned out to be parasitized (fig1e), rather than unparasitized hosts that might simply be trying to escape parasitism. This is a stronger clue that this may be for “superparasitism avoidance” than the observation that female parasitoids prefer to search on the medium; the latter could simply indicate that larvae try to escape from being attacked by a parasitoid.

After discussion with our colleagues, we keep Fig.1e as its original position, but accordingly reorganized this paragraph to propose both two explanations (see lines 104-106).

L 182 – 183: “most tissues”? It is unclear from the methods section which other tissues were dissected or examined.

We have provided related details in the Method section (Lines 779-781).

L 218 – 220: this experiment is a very important part to show the causality of EsGAP in inducing ROS in the CNS, which then relates to the hosts’ induced behaviour. However, it is only presented with a qualitative description, and not with a statistical analysis. This part of the study requires more details and a substantive analysis of the patterns.

To address your concerns, we have newly added two aspects of works. First, we have quantified the difference by analyzing the local stained spots, which showed statistically significant difference (see lines 228-230 and newly added Fig. 3j). Furthermore, we have generated the transgenic flies that specifically express EsGAP1 proteins of parasitoids in CNS, with the help of *Elav-GAL4*. These flies were found significantly higher ROS level than control flies, again supporting the causality of EsGAP in inducing ROS in CNS (see lines 230-233 and newly added Figs. 3k & 3l).

L 240 – 243: I believe CK means “not exposed to parasitoids”, so for each species, the numbers of eggs in the various species is not “in CK” but “for parasitoids with the control construct P_{dsGFP}”

We have changed “CK” to “P_{dsGFP}” to indicate the parasitization by the parasitoids with the injection of control *dsGFP*.

L250: not clear how these results “strongly indicate” a similar role in other drosophilid parasitoids: they indicate that Lb can manipulate this behaviour in a range of host species.

The transition here has been reorganized according to contexts (see line 266-268).

L253: replace “relative” with “closely related”?

It has been replaced as suggested.

L 258: replace “modes” with “clades”?

We have replaced replace “modes” with “clades”.

L292: could this speculation on “LGT from protists” not be tested with a fairly simple phylogenetic analysis/comparison, by expanding the analysis for figure 5a with protists/prokaryotes?

Our original Fig. 5a did include protist species, i.e. those non-metazoan eukaryotes. However, BLASTP against NR indicates that all hits are from Eukaryotes. Note that the original Fig. 5a has been moved to Supplementary Fig. 7.

L 313: should these results not be presented in the main text as part of figure 5? They are crucial to show that the host behaviour manipulation is also found in a related parasitoid species, and not an unique feature of *L. boulandi*.

This is a great idea. We have presented these results in new Fig. 5b.

L 321: I do not agree that it has been a “concern”: clear benefits and mechanisms have been proposed, such as the ability for host discrimination.

We apologize for the confusion. We meant to express that this question is long concerned. This sentence has been reworded accordingly (see line 344).

L 321 – 325: I suggest merging these two sentences on the ability of parasitoids for host discrimination into one

We have reorganized the content around here (see lines 344-347).

L 325: “infectious information” is not clear or accurate (the larvae are not infectious)

We have changed this as “parasitized status”.

L 331 – 342: this entire section is confusing and partially incorrect. “Host discrimination” is not a real contrast to the “host behaviour manipulation” that is investigated in this study: host discrimination is a strategy that females may employ when they encounter already parasitized hosts (or unsuitable host species). The manipulation of host behaviour is a strategy that females may employ to reduce the risk of other females “finding” the host they have already exploited. The benefits of this strategy of host behaviour manipulation are not primarily for the female that may waste time or eggs by superparasitism, but for the female that has already made an actual investment in this host (deposited an egg in it), and can so reduce the risk of other females

exploiting this host as well (by superparasitization).

Thank you again for helping us clarify the ecological consequence of superparasitism avoidance. We have reorganized this section based on your inputs here and above (see lines 353-366).

L 346: consider to present Figure 5c as a separate figure. It is not connected to figure 5a and 5b, and would benefit from a more extensive legend.

We have presented this proposed model to a separated figure (Fig. 6) and substantially extended the legend (see lines 557-565).

L361 - 364: I would consider it much more likely that these venom proteins are there to stop the host from differentiating lamellocytes (i.e., to try to sabotage the hosts' immune response), than to avoid damage to host neurons.

Thank you for reminding us of the potential relationship of SOD with suppressing the immune response of hosts. Our original link with host neurons was based on the broadly reported role of SOD in controlling ROS. We still think this relationship stands and have mentioned the alternative role as you pointed out in main text (see lines 391-395).

L391: what does "this strategy" refer to?

This strategy refers to the acquisition of a beneficial function of RhoGap domain. Here we have explained it out to avoid confusion (see line 435-436).

L 396 – 398: Too broad statement. Competition can also lead to strong specialization or a reduced realized niche (compared to the fundamental niche). Its relation with host range expansion has no good foundation.

We acknowledge the alternative consequences of competition as you pointed out. However, the "statements" regarding evolutionary diversification are based on previous studies but our data. We think this consequence is related to our observed "attempts" of the specialist Lb in other hosts. This paragraph has been reorganized, in which we only put the observation together with the referenced statement and let audiences judge the potential link by themselves (see lines 435-450).

L 402: "the recent study" is a bit confusing. It is not the current study, but a related study? I also do not fully understand this statement.

This is a recently published study that reported the divergent adaptation strategies between Lb and Lh. We have changed "the recent study" to "a related study".

L 406: remove the part “that can be recognized by the wasp”. The most parsimonious explanation is that it “induces behaviour in the host to limit [the risk of] superparasitism”.

We have modified this sentence as suggested (see line 453).

L 410 – 411: I fail to see how this study “provided key empirical evidence for how [...] it structures ecological communities at long time scales”. Suggest removal of the latter part of the sentence, or to provide clarification.

This sentence has been removed.

L435: please, rephrase and explain “Genes are displayed based on the TPM of venom”.

Genes were ordered based on the rank of expression levels of the venom. We have rephrased the sentence as “Genes are displayed in order of expression levels in the venom.”

L 448 & L 453: “The dashed line” instead of “Dashed lines”?

These have been modified as suggested.

Figure 4a: this figure suggests that the number of hosts that were “without eggs” was also higher in PdsEsGAP1? Did you test for this?

Indeed, the nonparasitized ratio of hosts got increased upon the exposure to EsGAP1-interference parasitoids. We have newly added a supplementary figure (Supplementary Fig. 10) to independently show the difference and statistics. Under the condition of less parasitoid hosts escaping, parasitoids have a higher probability to parasitize a parasitized host. It thus makes sense that the distribution of three potential outcomes becomes more even, in comparison to the condition that parasitized hosts being driven away. We have mentioned these results in the revised manuscript (lines 362-366).

Legend Figure 5a: please explain both the circle and asterisk in the legend of the figure

In the original manuscript, the asterisk indicated the lineage-specific lineage while the circle indicated typical orthologous lineage. They had been mentioned in main text of original manuscript but accidentally neglected in the legend. In the revised figure (new Supplementary Fig. 7), these symbols have been replaced by direct labels.

L 497- 500 / Legend Figure 5c: I believe the proposed model is that this is a strategy, not to combat their host, but to “avoid superparasitism (or competition) by conspecifics. Replace “leading to” by “inducing”, and I suggest removing “and subsequent superparasitism avoidance”. The sentence on long-term effects, host shift or expansion or speciation seem not warranted or directly related to the findings.

The long-term effects, e.g. host range expansion and speciation, are our proposed potential ecological consequence, which were indicated by “dashed” arrows. We have removed “speciation” and “host shift” due to lacking of supports, but retained a further toning down “host expansion”. The potential link has been further explained by extended discussion (lines 435-450).

L 506: too little information to repeat the rearing conditions (e.g. quantities and % of apple juice agar)

We have provided more detailed information of rearing conditions in Methods (see lines 568-576).

L 512: How many parasitoid females were simultaneously released on a host patch?

We apologize for the incomplete information. The exact numbers of hosts for different assays have been provided in legends of corresponding figures and source data. The number of released wasps is available based on the host/wasp ratio and further provided in source data.

L 512 L518: Replace “Well-mated” with “Mated”: females are either mated or not

“Well” has been removed.

L 519: what is the “immediate repellency behaviour” and is this similar to the observed escape behaviour after parasitism?

We note that there is an immediately avoidance behaviour of *Drosophila* larvae upon the release of wasps, which is similar to the focused escape behavior of this study. However, this immediate escape behaviour only occurs before the anesthetization relief of wasps and lasts less than 2 min. By contrast, the focused escape behaviours of this study are all observed after released parasitoids being completely recovered from anaesthesia and ready for parasitization. Although this immediate response does not overlap with our focused behaviors at all, we still hope to mention this observed phenotype to avoid causing confusion for other researchers who might repeat our behavioral assays. Detailed explanation has been added in main text (see lines 604-609).

L 524: how were the escape indices calculated?

The escape indices were calculated as the percentage of the number of hosts exhibiting escape behavior to the number of total hosts. Related information has been added in Methods (see lines 615-616).

L 531 – 547: consider moving this section to later, to maintain the same order as in the results.

This section has been moved to later accordingly.

L540: Replace “Figure 2b and 2d” with “Figure 3”

These typos have been fixed.

L572: provide details on the collection of the different life stages and venom for transcriptome analysis

Related details have been provided (see lines 658-669).

L 587 & L589: write out the abbreviations KO and TMP

Both KO and TMP have been spelled out.

L 601: I am no expert on protein analysis, but why was only the supernatant analyzed and not the pellet?

The proteins with SDT lysis buffer treatment and/or with the boiling treatment are denatured, and dissolved in the buffer. Thus, the supernatant, but the pellet, contains the portion of the pure protein.

L 634: Rephrase to something like “Genes in the venom with an expression of 7.3 transcripts per million (TPM) were”

We have rephrased the sentence as suggested. Note that TPM has been spelled out above.

L 635 - 637: Rephrase and clarify

This is actually a threshold to strictly define venom proteins based on both transcriptome and proteome evidence. The sentence has been reorganized as “VG-expressed genes that were able to be fully aligned to at least three proteomic peptides and were ultimately defined as venom proteins”.

L 686: Clarify “Reactions”. Is this including the cDNA synthesis? If not, please provide these details.

The reactions here include both the cDNA synthesis and the qPCR. We have modified it accordingly (see line 771).

REVIEWERS' COMMENTS

Reviewer #2 (Remarks to the Author):

The authors thoroughly addressed all of my concerns. I continue to believe that this is a strong paper that will be of general interest to the readers of Nature Communications.

Reviewer #3 (Remarks to the Author):

The authors have considerably revised the manuscript, to address all the queries and comments of the reviewers. The series of experiments they have done are very impressive, and reveal a fascinating strategy of Leptopilina parasitoids, including the proximate mechanisms and evolutionary history. I strongly believe that this study is of high interest to the readership of Nature Communications, and of very high quality. I commend the authors on the comprehensive series of experiments, leading to a very strong and convincing manuscript. At places, the presentation of the context is still a bit unclear, or not so well formulated, weakening the argumentation. I provided a number of suggestions (minor comments), but this is only in an attempt to help the authors improve the clarity of their text.

I will restrict my comments to my main concerns in the previous version of the manuscript:

The introduction has been narrowed, to better fit with the intraspecific competition context of the superparasitism avoidance. Most of the concerns I raised on interpretation have been properly addressed, and this has much improved the theoretical embedding of the research. Also in the discussion, the process of superparasitism and the evolution of strategies to avoid it, are presented with much more clarity. I thank the authors for so carefully redrafting these parts. There are still a few statements that are slightly off or unclear, in my opinion (see below for details). Also, parts of the manuscript could benefit from language editing by a native speaker (see below for some suggestions). All these corrections are really minor, and do not diminish from the high value of this study.

Concerning the missing details of the methods section and the statistical analysis: I feel these have been fully addressed in the revised manuscript.

Also the rearrangements of the first part, including the reorganisation of Figure 1 has improved clarity.

Minor comments:

L24: remove “better” (adaptations are inherently “better”)

L28-30: Consider rephrasing to: “Here, we show that the solitary endoparasitic wasp *Leptopilina boulardi* provokes an escape response in its *Drosophila* host for superparasitism avoidance. “

L48-50: Unclear formulation, as wasp offspring cannot switch hosts, but it is the strategy of ovipositing wasp females that is under selection. Consider rephrasing to something like: Furthermore, the offspring of parasitic wasps are typically confined to the host that was parasitized, so the strategies of the ovipositing female therefore largely determine her reproductive outputs; her strategy is therefore under strong natural selection.

L51: first use of the term “superparasitism”, but it is defined only later (L56)

L61: add “being” before considered

L81-82: needs some rephrasing, e.g. Even WHEN multiple ovipositions occur, ...

L82: Typo in behavioural / behavioral

L82-84: unclear sentence. Do you mean “defined” rather than “definite”. And is this a behavioural qualification?

L90: rephrase “most larvae parasitized a single egg for up to 60 min”, for example to: “most larvae contained a single (parasitoid) egg after 60 min”

L99: “Upon the release of parasitoids” suggests that the observed behaviour follows directly after the release of the parasitoids. I think the pattern was observed over a longer time frame during the experiment? If so, I would suggest to rephrase this slightly to avoid confusion.

L103-105: Although the addition (L 104-106 “or simply indicates ... parasitized) is good, the first part of this argumentation is not formulated well: it is only in the larvae’s interest to “avoid further parasitism” if double parasitism further increases their own risk of death – if that is the argument, this explanation is missing; However, this is not so much a contrast to the second, more simple explanation (avoiding attack by a parasitoid). I believe what the authors want to postulate here is something like: “such escape behaviour may have been induced by parasitism, by a parasitoid that tries to manipulate the hosts’ behaviour to avoid superparasitism.

L120-123: Rephrase sentence somewhat, for example to: In contrast, the non-escaped *Drosophila* larvae contained many more eggs than the ones that had shown escape behaviour (xxx), supporting that eventually the non-escaping hosts suffered higher levels of superparasitism.

L155: “significantly highly expressed” is unclear: is it perhaps “significantly higher expressed in the venom than in whole-body samples of eggs, larvae, pupae and adults”

L243-244: rephrase sentence part “where each (Fig 4a,b)” , for example to “and that *D. melanogaster* hosts treated with PdsEsGAP1 had approximately 1.37 eggs, which is an increase of 34% in comparison to the PdsGFP-treated control hosts (Fig 4a,b)”

L257: rephrase: PdsEsGAP lead to a significant increase in superparasitism in all tested host species (Fig. 4a), with on average ~1.67 eggs in *D. hydei* (increased 49% in comparison to control),” etc

L268-270: the conclusion that orthologues were not found does not seem to match with the information shown in Figure 5a. The later text does refer to the clades in Lb and Lh, but it is not clear from the text why the orthologues were not found?

L170: I would suggest replacing “parasite” with “parasitoid”.

L281: the description “genes within the orthologue sublineage” is not entirely clear. Do the authors mean the “lineage containing one-to-one orthologues”? The following text is also difficult to follow, as the descriptions of the sublineages is not entirely unambiguous. Perhaps it would be an idea to label the sublineages as I and sublineage II (and maybe even sublineage III if the top clade is also considered as a separate clade), also in the figure 5a, to avoid unclarity.

L331: suggest replacing “despite” with “albeit with a similar non-specific xxx”

L338: I would suggest deleting “i.e. i.e. host shift among *Leptopilina* species”, as it is unclear and not entirely accurate, as two species share a number of host species. While *L. heterotoma* has a more extensive range (and is therefore a generalist), it is not clear whether *L. heterotoma* expanded its host range, or whether *L. bouvardi* restricted its host range (became a specialist). Therefore referring to this difference as “host shift” is a bit unclear.

L344: replace “concerned” with something else, e.g. a topic of study, or pondered

L346: Suggest deleting “, and that” and start a new sentence with “Host discrimination”

L358: suggest replacing “reject” with “reduce encounters of”

L364: replace “than” by “compared to”.

L365-366: Sentence unclear. Replace by, for example, “This further supports that a superparasitism avoidance strategy enhances the efficiency of parasitoids”.

L389: replace “much” by “very”

L408L: replace “are” by “is”

L435: ecological speciation?

L439: replace “colonize” with “utilize”?

L445: replace “occasional” with “exceptional”?

Bregje Wertheim

REVIEWERS' COMMENTS

Reviewer #2 (Remarks to the Author):

The authors thoroughly addressed all of my concerns. I continue to believe that this is a strong paper that will be of general interest to the readers of Nature Communications.

We appreciate your high evaluation and inputs that help improve the study.

Reviewer #3 (Remarks to the Author):

The authors have considerably revised the manuscript, to address all the queries and comments of the reviewers. The series of experiments they have done are very impressive, and reveal a fascinating strategy of Leptopilina parasitoids, including the proximate mechanisms and evolutionary history. I strongly believe that this study is of high interest to the readership of Nature Communications, and of very high quality. I commend the authors on the comprehensive series of experiments, leading to a very strong and convincing manuscript. At places, the presentation of the context is still a bit unclear, or not so well formulated, weakening the argumentation. I provided a number of suggestions (minor comments), but this is only in an attempt to help the authors improve the clarity of their text.

We appreciate your high evaluation and inputs that help improve the study.

I will restrict my comments to my main concerns in the previous version of the manuscript:

The introduction has been narrowed, to better fit with the intraspecific competition context of the superparasitism avoidance. Most of the concerns I raised on interpretation have been properly addressed, and this has much improved the theoretical embedding of the research. Also in the discussion, the process of superparasitism and the evolution of strategies to avoid it, are presented with much more clarity. I thank the authors for so carefully redrafting these parts. There are still a few statements that are slightly off or unclear, in my opinion (see below for details). Also, parts of the manuscript could benefit from language editing by a native speaker (see below for some suggestions). All these corrections are really minor, and do not diminish from the high value of this study.

Thank you for the kindly help with language editing. We have carefully addressed your concerns as below.

Concerning the missing details of the methods section and the statistical analysis: I feel these have been fully addressed in the revised manuscript.

Also the rearrangements of the first part, including the reorganisation of Figure 1 has improved clarity.

Thank you.

Minor comments:

L24: remove “better” (adaptations are inherently “better”)

It has been corrected as suggested.

L28-30: Consider rephrasing to: “Here, we show that the solitary endoparasitic wasp *Leptopilina boulardi* provokes an escape response in its *Drosophila* host for superparasitism avoidance. “

It has been corrected as suggested.

L48-50: Unclear formulation, as wasp offspring cannot switch hosts, but it is the strategy of ovipositing wasp females that is under selection. Consider rephrasing to something like: Furthermore, the offspring of parasitic wasps are typically confined to the host that was parasitized, so the strategies of the ovipositing female therefore largely determine her reproductive outputs; her strategy is therefore under strong natural selection.

It has been corrected as suggested.

L51: first use of the term “superparasitism”, but it is defined only later (L56)

It has been corrected as suggested.

L61: add “being” before considered

It has been corrected as suggested.

L81-82: needs some rephrasing, e.g. Even WHEN multiple ovipositions occur, ...

It has been corrected as suggested.

L82: Typo in behavioural / behavioral

It has been corrected as suggested.

L82-84: unclear sentence. Do you mean “defined” rather than “definite”. And is this a behavioural qualification?

Here we meant to express that determining superparasitism is easier and clearer in solitary parasitoids, because at most one of the laid offsprings can successfully develop into the adult and emerge from a single host (if survived). Thus, solitary parasitoids provide appropriate model to study superparasitism behavior. To avoid confusion, we have changed “definite” to “unambiguous”.

L90: rephrase “most larvae parasitized a single egg for up to 60 min”, for example to: “most larvae contained a single (parasitoid) egg after 60 min”

It has been corrected as suggested.

L99: “Upon the release of parasitoids” suggests that the observed behaviour follows directly after the release of the parasitoids. I think the pattern was observed over a longer time frame during the experiment? If so, I would suggest to rephrase this slightly to avoid confusion.

We have changed “upon” to “after”.

L103-105: Although the addition (L 104-106 “or simply indicates ... parasitized) is good, the first part of this argumentation is not formulated well: it is only in the larvae’s interest to “avoid further parasitism” if double parasitism further increases their own risk of death – if that is the argument, this explanation is missing; However, this is not so much a contrast to the second, more simple explanation (avoiding attack by a parasitoid). I believe what the authors want to postulate here is something like: “such escape behaviour may have been induced by parasitism, by a parasitoid that tries to manipulate the hosts’ behaviour to avoid superparasitism.

It has been corrected as suggested.

L120-123: Rephrase sentence somewhat, for example to: In contrast, the non-escaped *Drosophila* larvae contained many more eggs than the ones that had shown escape behaviour (xxx), supporting that eventually the non-escaping hosts suffered higher levels of superparasitism.

Thank you. It has been corrected as suggested.

L155: “significantly highly expressed” is unclear: is it perhaps “significantly higher expressed in the venom than in whole-body samples of eggs, larvae, pupae and adults”

We performed Z statistics to identify genes of particularly high level of expression that are deviated from the overall distribution. This is not the comparison between tissues. We have used “extremely” instead.

L243-244: rephrase sentence part “where each (Fig 4a,b)” , for example to “and that *D. melanogaster* hosts treated with PdsEsGAP1 had approximately 1.37 eggs, which is an increase of 34% in comparison to the PdsGFP-treated control hosts (Fig 4a,b)”

It has been corrected as suggested.

L257: rephrase: PdsEsGAP lead to a significant increase in superparasitism in all tested host species (Fig. 4a), with on average ~1.67 eggs in *D. hydei* (increased 49% in comparison to control),” etc

They have been corrected as suggested.

L268-270: the conclusion that orthologues were not found does not seem to match with the information shown in Figure 5a. The later text does refer to the clades in Lb and Lh, but it is not clear from the text why the orthologues were not found?

All EsGAPs of Lb and Lh are clustered respectively, rather than the 1vs1 orthologous across species as other rhoGAP genes.

L270: I would suggest replacing “parasite” with “parasitoid”.

It has been corrected as suggested.

L281: the description “genes within the orthologue sublineage” is not entirely clear. Do the authors mean the “lineage containing one-to-one orthologues”? The following text is also difficult to follow, as the descriptions of the sublineages is not entirely unambiguous. Perhaps it would be an idea to label the sublineages as I and sublineage II (and maybe even sublineage III if the top clade is also considered as a separate clade), also in the figure 5a, to avoid unclarity.

We have changed “orthologue sublineage” to “one-to-one orthologues” and indicated them as “non-shadowed” and “green shadowed” in main text, respectively.

L331: suggest replacing “despite” with “albeit with a similar non-specific xxx”

It has been corrected as suggested.

L338: I would suggest deleting “i.e. i.e. host shift among Leptopilina species”, as it is unclear and not entirely accurate, as two species share a number of host species. While *L. heterotoma* has a more extensive range (and is therefore a generalist), it is not clear whether *L. heterotoma* expanded its host range, or whether *L. boulardi* restricted its host range (became a specialist). Therefore referring to this difference as “host shift” is a bit unclear.

This place has been changed to “... prior to the divergence between *Lb* and *Lh* that show differences in host range”. Their difference in host range is clear.

L344: replace “concerned” with something else, e.g. a topic of study, or pondered

It has been corrected to “pondered”.

L346: Suggest deleting “, and that” and start a new sentence with “Host discrimination”

It has been corrected as suggested.

L358: suggest replacing “reject” with “reduce encounters of”

It has been corrected as suggested.

L364: replace “than” by “compared to”.

It has been corrected as suggested.

L365-366: Sentence unclear. Replace by, for example, “This further supports that a superparasitism avoidance strategy enhances the efficiency of parasitoids”.

It has been corrected as suggested.

L389: replace “much” by “very”

It has been corrected as suggested.

L408: replace “are” by “is”

It has been corrected as suggested.

L435: ecological speciation?

We have changed to "... one of the main ecological forces in driving speciation".

L439: replace "colonize" with "utilize"?

It has been corrected as suggested.

L445: replace "occasional" with "exceptional"?

It has been corrected as suggested.